# Electrospun Scaffolds of Polylactic Acid, Collagen, and Amorphous Calcium Phosphate for Bone Repair

**DOI:** 10.3390/pharmaceutics15112529

**Published:** 2023-10-25

**Authors:** William Cárdenas-Aguazaco, Bernardo Camacho, Edwin Yesid Gómez-Pachón, Adriana Lorena Lara-Bertrand, Ingrid Silva-Cote

**Affiliations:** 1Instituto Distrital de Ciencia, Biotecnología e Innovación en Salud-IDCBIS, Bogotá 111611, Colombia; wcardenas@idcbis.org.co (W.C.-A.); bacamacho@idcbis.org.co (B.C.); alara@idcbis.org.co (A.L.L.-B.); 2Facultad Duitama, Universidad Pedagógica y Tecnológica de Colombia-UPTC, Duitama 150462, Colombia; edwin.gomez02@uptc.edu.co

**Keywords:** electrospinning, bone repair, PLA, calcium phosphate, hydroxyapatite

## Abstract

Most electrospun scaffolds for bone tissue engineering typically use hydroxyapatite (HA) or beta tricalcium phosphate (β-TCP). However, the biological activity of these crystalline compounds can be limited due to their low solubility. Therefore, amorphous calcium phosphate (ACP) may be an alternative in bone repair scaffolds. This study analyzes the morphology, porosity, mechanical strength, and surface chemistry of electrospun scaffolds composed of polylactic acid and collagen integrated with hydroxyapatite (MHAP) or amorphous calcium phosphate (MACP). In addition, the in vitro biocompatibility, osteogenic differentiation, and growth factor production associated with bone repair using human Wharton’s jelly-derived mesenchymal stem cells (hWJ-MSCs) are evaluated. The results show that the electrospun MHAP and MACP scaffolds exhibit a fibrous morphology with interconnected pores. Both scaffolds exhibit favorable biocompatibility and stimulate the proliferation and osteogenesis of hWJ-MSCs. However, cell adhesion and osteocalcin production are greater in the MACP scaffold compared to the MHAP scaffold. In addition, the MACP scaffold shows significant production of bone-repair-related growth factors such as transforming growth factor-beta 1 (TGF-β1), providing a solid basis for its use in bone tissue engineering.

## 1. Introduction

Bone repair is a major challenge in tissue engineering due to the high incidence of fractures and bone defects caused by trauma, disease, and non-union fractures [1,2]. Researchers have developed methods in tissue engineering that use biomaterials, cells, and bioactive substances to address these challenges [3]. Biomaterials such as natural or synthetic polymers, inorganic composites, and hybrid blends are critical for the fabrication of bone scaffolds that provide mechanical stability, biocompatibility, biodegradability, osteoconductivity, and osteoinductivity [4,5].

Polylactic acid (PLA) is a biodegradable and biocompatible synthetic polymer widely used in biomedical applications for bone tissue due to its ease of processing, mechanical stability, and versatility [6]. PLA has some notable drawbacks, such as low hydrophilicity, slow degradation, and limited bioactivity [7]. To overcome these limitations, PLA can be combined with other biomaterials, including collagen (COL) and calcium phosphates (CaPs). COL, a natural protein found in the extracellular matrix of bone, has high biocompatibility, bioactivity, and hydrophilicity [8]. Meanwhile, bioceramics known as CaPs have become a common choice for bone repair systems. Their excellent biocompatibility, osteoconductivity, and similarity to the natural bone mineral phase have made them a widely adopted solution [9]. CaPs can be classified based on their chemical composition, crystal structure, and solubility; they include HAP, β-TCP, octacalcium phosphate (OCP), and ACP [10]. Each type of CaP has advantages and disadvantages for bone regeneration applications depending on specific properties such as mechanical strength, degradation rate, and bioactivity [11]. HAP is the CaP that is more frequently used in bone tissue engineering because it is the primary inorganic component of bone mineral. This bioceramic possesses remarkable osteoinductive and osteoconductive properties [12]. Furthermore, research has demonstrated that combining it with PLA and COL to fabricate scaffolds is a feasible option for bone tissue engineering [13].

On the other hand, there are several techniques to create scaffolds by combining these three materials. One such technique is electrospinning, a versatile and reproducible method that uses an electric field on a polymer solution to produce micro or nanofibers [14]. Electrospun scaffolds have high porosity and a large specific surface area, and they incorporate the properties of each material, facilitating cellular interaction and the infiltration of nutrients and growth factors [15]. In addition, electrospinning allows for control over the fiber diameter, orientation, composition, and incorporation of bioactive agents [16].

Several studies have reported on the fabrication and characterization of electrospun PLA, COL, and HAP scaffolds for bone applications. These studies showed promising results regarding morphology, composition, mechanical properties, and biological performance [17,18]. However, HAP’s high crystallinity and stoichiometry can result in a relatively slow dissolution rate, which could negatively affect bone repair processes [19]. An alternative to HAP is ACP; due to its disordered structure and structural defects, ACP is a precursor phase of HAP that exhibits higher reactivity, solubility, and bioactivity [20]. ACP can release calcium and phosphate ions in a physiological context. This phenomenon promotes the growth of HAP crystals on the scaffold’s surface and enhances the bone cell differentiation process towards osteogenesis [21].

Although few studies have investigated the integration of ACP into electrospun scaffolds for bone applications [22,23], these scaffolds have shown that they are promising for bone tissue engineering and have the potential to promote bone regeneration due to their rapid mineralization and improved cell adhesion [24,25,26]. The quantity and evenness of ACP dispersion in the electrospun fibers have an impact on the morphology, stability, and functional traits of the resulting scaffolds. The overabundance or deficiency of ACP could potentially compromise both the mechanical and bioactive properties of these scaffolds. In addition, the physicochemical properties of electrospun scaffolds incorporating ACP in conjunction with proteins such as collagen have not been studied, nor have their effects on primary cells been evaluated.

The hWJ-MSCs are primary cells with a high proliferative capacity, low immunogenicity, and ability to differentiate into osteoblasts, chondrocytes, and adipocytes. Previous studies have shown the potential of hWJ-MSCs for tissue regeneration, particularly in cartilage and bone. These cells possess immunoprivileged or hypo-immunogenic properties [27,28], which make them a viable option for allogeneic therapies. Additionally, hWJ-MSCs generate paracrine factors and immunomodulators that aid in the recruitment of other cells for tissue repair [29].

To better comprehend the effects of ACP on the physicochemical characteristics of electrospun scaffolds made of PLA and COL, and their implications for bone regeneration, we created and analyzed the physicochemical properties, as well as evaluated the biological properties of electrospun scaffolds incorporating PLA and COL alongside HAP or ACP, labeled MHAP and MACP. The aim is to investigate the impact of bioceramics on scaffold properties and their subsequent influence on the behavior of hWJ-MSCs in regenerative medicine studies. 

## 2. Materials and Methods

### 2.1. Materials

Calcium hydroxide (Ca (OH)_2_, 95% purity), orthophosphoric acid (H_3_PO_4_, 85% purity), ammonium hydroxide solution (NH_4_OH, 28–30%), HAP (<200 nm particle size), and COL (from calf skin type I) were acquired from Sigma-Aldrich (St Louis, Missouri, USA). PLA (Mw 90 kDa) was acquired from 3DBOTS, and the solvent, 2, 2, 2 trifluoroethanol (TFE, 99% purity), was acquired from Merck Millipore (Darmstamdt, Hesse, Germany). 

### 2.2. Synthesis of Amorphous Calcium Phosphate

A wet chemical precipitation reaction was used to obtain ACP [30]. A total of 79.55 g of Ca(OH)_2_ was dissolved in 500 mL of distilled water at room temperature. Then, 97.35 g of H_3_PO_4_ was added at 1.5 mL/min. This solution remained at room temperature for 24 h while stirring continuously. Then, 10 g of NH_4_OH was added drop by drop until the solution was brought to a pH level of 10, in order to generate the precipitation of the solid. This precipitate was filtered under a vacuum and dried at 100 °C for one hour. 

### 2.3. HAP and ACP Characterization

The morphology, size, and calcium/phosphorus ratio of HAP and the synthesized ACP were examined using scanning electron microscopy (SEM) and X-ray dispersive spectroscopy (EDX). To accomplish this, a microscope (JSM-6510, JEOL, Tokyo, Japan) equipped with an EDX detector (INCA Energy, Oxford Instruments, Abingdon, UK) was utilized. Images were captured at various magnifications, and the average diameter of the compounds was determined using ImageJ software (https://imagej.net/software/fiji/downloads (accessed on 27 July 2023)). EDX spectra were also obtained in the 0–10 keV range, with a resolution of 10 eV and an acquisition time of 60 s. The INCA software (https://www.etas.com/en/products/inca_software_products.php (accessed on 24 July 2023)) was employed to calculate the atomic and weight percentages of the detected elements.

XRD data were collected using a high-resolution X-ray diffractometer PANalytical from the X’PERT PRO line (Malvern Panalytical, Malvern, UK) equipped with a copper (Cu) X-ray source (λ = 1.5406 Å) operating at 40 kV and 30 mA. Data were collected using a scintillation detector with a 2-s counting time per step, over a 2θ range of 20° to 50° and a step size of 0.02°. The diffraction patterns obtained from the sample underwent processing to yield 2θ vs. intensity plots.

### 2.4. Electrospun Scaffold Fabrication

MHAP and MACP solutions were mixed at a weight ratio of 10:1:1 (PLA:COL:CaP) according to the ratios proposed by Akkouch et al. [31], with some modifications (see Table 1). Additionally, a control scaffold of neat PLA was used. The materials were homogenized in TFE at 500 rpm and 30 °C for a duration of 3 h. Electrospinning was then conducted at 20 °C using a square horizontal static collector measuring 5 cm × 5 cm, following the parameters outlined by Gomez-Pachón et al. [32], with some modifications (15 kV, 15 cm, and 2 mL/h). The resulting scaffolds were placed in a desiccator until they were ready to be used for physicochemical characterization or cellular assays.

### 2.5. Scaffold Characterization

A Fourier transform infrared spectroscopy (FTIR) analysis was conducted to identify the functional groups present in the scaffolds. For this purpose, a spectrophotometer (Nicolet 510P, Thermo Scientific, Waltham, MA, USA) equipped with an attenuated total reflectance (ATR) accessory was utilized. Spectra were recorded in the range of 4000–500 cm^−1^, with a resolution of 4 cm^−1^ and 32 scans.

The elemental composition of the scaffolds was determined through X-ray photoelectron spectroscopy (XPS). Measurements were performed using a spectrometer (VERSAPROB II, Physical Electronics GmbH, Feldkirchen, Bavaria, Germany) with a 200 W Mg X-ray source (h = 1253.6 eV) and a 50 eV hemispherical analyzer. Spectra were recorded in the range of 0–1100 eV, with a resolution of 0.1 eV and a step of 0.05 eV. To correct the charge shift, the C1s peak at 284.8 eV was used as an internal reference.

The diameters, porosity, and morphology of the electrospun fibers were observed using SEM (JEOL-JSM-7600F, Tokyo, Japan). Prior to analysis, the scaffolds were coated with gold using a sputter coater. From the obtained images, the average diameter of the fibers was determined using ImageJ software (https://imagej.net/software/fiji/downloads (accessed on 2 February 2023)).

The glass transition temperature (Tg), crystallization temperature (Tc), and melting temperature (Tm) of the electrospun scaffolds were determined through a calorimetric analysis (DSC). A calorimeter (DSC Q100, TA Instrument, New Castle, DE, USA) with nitrogen as the purge gas was employed. Approximately 5 mg of each sample was weighed and subjected to a heating ramp from 20 °C to 250 °C, with a heating rate of 10 °C/min.

Finally, the tensile test was performed following ASTM D1708 standards using an INSTRON 1125 (INSTRON, Norwood, MA, USA) universal uniaxial tensile testing machine. The test was conducted at a speed of 10 mm/min with a load of 5 kN. Five specimens were tested to calculate the Young’s modulus of the scaffolds. All procedures described above were carried out at room temperature.

### 2.6. Cytotoxicity Assay

The in vitro cytotoxicity test was performed as a direct contact assay in accordance with ISO 10993-5 [33]. The hWJ-MSCs were seeded at a density of 3 × 10^4^ cells/well in 24-well plates and cultured in Dulbecco’s Modified Eagle Medium (DMEM) supplemented with human platelet lysate (hPL, 10% *v*/*v*), 100 IU/mL penicillin, 2.5 μg/mL streptomycin, and 80 μL heparin and incubated at 37 °C in a humidified atmosphere containing 5% CO_2_ for 24 h. After 24 h of cell culture, the scaffolds were carefully placed on the cell monolayer in the center of each of the replicate wells and incubated for 24 and 48 h [34]. The culture medium was then removed, and resazurin was introduced to determine the cytotoxic effects. The redox dye resazurin (0.01 mg/mL, Sigma-Aldrich, St. Louis, MO, USA) was added to each well at a concentration of 1% (*v*/*v*). The wells were then incubated for three hours at 37 °C in an atmosphere that contained 5% CO_2_ and was protected from light. After incubation, the supernatant was collected and transferred to a 96-well plate. The Synergy MX spectrophotometer microplate reader (Biotek Synergy, Winooski, VT, USA) was used to measure the reduction of resazurin (non-fluorescent blue dye) to resorufin (pink fluorescent) at an excitation wavelength of 570 nm and an emission wavelength of 600 nm [35]. Three cell donors were used, and three replicates were performed for each condition.

### 2.7. Cell Adhesion and Proliferation

To assess the effects of the electrospun scaffolds on cell adhesion and proliferation, we utilized hWJ-MSCs that were transduced with green fluorescent protein (hWJ-MSCs-GFP). The scaffolds were cut into circular discs 1 cm in diameter and plated in 48-well plates. Subsequently, the scaffold discs were sterilized using the BIOBEAM 2000 caesium-137 gamma irradiation device (Eckert & Ziegler BEBIG GmbH, Berlin, Germany). Two doses of 25 Gray each were administered at room temperature for 15 min under light-protected conditions. After sterilization, a cell density of 5 × 10^4^ cells was seeded in Dulbecco’s Modified Eagle Medium (DMEM) supplemented with human platelet lysate (hPL, 10% *v*/*v*), 100 IU/mL penicillin, 2.5 μg/mL streptomycin, and 80 μL heparin. The cells were incubated at 37 °C in a humidified atmosphere containing 5% CO_2_ for 24 h to facilitate cell adhesion. Following this, 400 microliters of the supplemented DMEM medium were added to the cells and changed every 2 to 3 days. The cells’ adhesion, morphology, and proliferation were observed using a Leica fluorescence microscope (DMi8-M). Images were captured with a Leica camera (Leica Microsystems GmbH, Wetzlar, Hesse, Germany) on days 1, 3, and 5.

In addition, cell organization on the scaffolds was evaluated after 7 days of culture with hWJ-MSCs by staining with phalloidin red (1:100 in PBS, Biolegend, San Diego, CA, USA) and 4′,6-diamidino-2-phenylindole dihydrochloride (DAPI, 0.1 ug mL^−1^ in deionized water, Thermo-Fisher Scientific), which allows precise visualization of the distribution of F-actin in the cell cytoskeleton and the location of nuclei, respectively. Images were captured using a confocal laser scanning microscope (Olympus FV1000, Tokyo, Japan) and analyzed with Fiji software (https://imagej.net/software/fiji/downloads (accessed on 11 August 2023)).

The proliferation of hWJ-MSCs on the PLA, MHAP, MACP, and TCP scaffolds was evaluated on days 1, 3, and 5 using the resazurin assay (Sigma-Aldrich). For each time point, 500 μL of a 1% resazurin solution in a fresh culture medium was added and incubated for 3 h after replacing the supplemented culture medium. After completion, 100 μL of supernatant from each treatment was collected, and the absorbance was recorded in a microplate reader (Synergy; BioTek, Winooski, VT, USA) at 570 nm and 600 nm [35]. A positive control group of cells seeded in tissue culture plate (TCP) was included.

### 2.8. Secretion of Growth Factors

The quantification of the platelet-derived growth factor (PDGFbb), basic fibroblast growth factor (bFGF), angiopoietin I, vascular endothelial growth factor (VEGF), and transforming growth factor-β1 (TGF-β1) secretion was performed using a magnetic bead-bound immunoassay (Luminex LXSAHM-08 from R&D Systems, Minneapolis, MN, USA) according to the manufacturer’s instructions. The Luminex plate was read on the Luminex^®^ Bio-Plex^®^ 200 system. In total, 5 × 10^4^ hWJ-MSCs from three donors were cultured on 1 cm diameter electrospun scaffolds (PLA, MHAP and MACP) under standard conditions. Supernatants were collected at 12, 24, 48, 72, and 96 h, centrifuged at 1200 rpm for 6 min to remove cellular debris, and stored at −20 °C. In addition, the protein concentration in the supplemented culture medium (0 h) and the supernatant of hWJ-MSCs seeded on tissue culture plates (TCP) were used as controls.

### 2.9. Osteogenic Differentiation

To induce osteogenic differentiation, hWJ-MSCs were seeded on the PLA, MHAP, and MACP scaffolds as previously described and treated with an osteogenic differentiation medium (StemPro Osteogenesis Differentiation Kit, Life Technologies, Carlsbad, CA, USA) 48 h after seeding. The medium was changed every 3 days until day 7. Calcium deposition was confirmed by staining sections of the samples with alizarin red (Sigma-Aldrich). Cells seeded on TCP were used as a control.

### 2.10. Immunohistochemical Analysis

Immunohistochemistry was performed on MHAP/hWJ-MSC and MACP/hWJ-MSC samples cultured in the supplemented DMEM and osteogenic differentiation medium to detect osteocalcin (OCN). Mouse monoclonal anti-OCN was used at 10 μg/mL (MAB1419, R&D Systems, Minneapolis, MN, USA). The procedure was performed according to the manufacturer’s instructions using the Vectastain Elite ABC Kit (Vector Laboratories, Burlingame, CA, USA). Briefly, deparaffinized and hydrated tissue sections were treated with urea tris buffer (pH 9.5) at 95 °C for 15 min to recover epitopes. Endogenous peroxidase activity was then blocked by treatment with 0.3% hydrogen peroxide in methyl alcohol for 10 min. The samples were incubated with the appropriate blocking serum (10% [*v*/*v*] goat, rabbit, or horse serum in phosphate-buffered saline (PBS), Vector Laboratories Vectastain Elite ABC Kit (Burlingame, CA, USA), followed by incubation with the primary antibody at 4 °C overnight. After three washes with PBS, sections were incubated with the appropriate biotinylated secondary antibody, followed by ABC peroxidase. The peroxidase reaction was developed using a substrate kit containing 3-3’-diaminobenzidine (Vector Laboratories, Newark, NJ, USA), and the reaction was terminated with distilled water.

### 2.11. SEM Imaging of Cells

After 7 days of culture in the supplemented DMEM and osteogenic differentiation medium, cells on the PLA, MHAP, and MACP scaffolds were fixed with 4% PFA (Sigma-Aldrich) for 1 h at 25 °C. The samples were rinsed three times with PBS and dehydrated in a series of alcohol solutions with increasing ethanol concentrations (50%, 70%, 90%, 96%, and 100%). They were then incubated in hexamethyldisilazane (HDMS, Sigma-Aldrich). After the complete evaporation of HDMS, the samples were coated with a gold layer and imaged by SEM.

### 2.12. Statistical Analysis

Statistical data analyses were conducted using ANOVA and Tukey’s post hoc test for multiple comparisons with GraphPad Prism version 8 software. In all instances, statistical significance was set at *p* < 0.05. All experiments were repeated three times.

## 3. Results

### 3.1. HAP and ACP Characterization

Figure 1a,b display SEM images of HAP and ACP, respectively. HAP presents a homogeneous particle size of 0.372 ± 0.196 µm and a regular, smooth surface. Conversely, amorphous calcium phosphate exhibits erratic aggregates measuring 3.212 ± 1.807 µm, together with a rougher and more porous texture. The discrepancy results from the structural differences between hydroxyapatite and amorphous calcium phosphate. Hydroxyapatite features a compact and orderly structure, whereas amorphous calcium phosphate has a less dense and disordered structure [36].

The EDX analysis evaluated the elemental composition of HAP and ACP, as shown in Figure 1c,d, respectively. The EDX spectra show the presence of calcium (Ca) and phosphorus (P) signals in both materials, confirming their identical elemental composition. Specifically, HAP shows atomic percentages of 3.37% for P and 5.50% for Ca, while ACP shows higher atomic percentages of 6.74% for P and 8.23% for Ca. However, the differences in the Ca/P ratio between the two compounds (1.63 for HAP and 1.22 for ACP) suggest variations in their degree of crystallinity. Based on the data presented by Munir et al. [37], it is clear that HAP is equivalent to calcium hydroxyapatite, while ACP is classified as amorphous calcium phosphate.

The XRD spectra, shown in Figure 1e, revealed distinct patterns for each material. In the HAP sample, three high-intensity peaks were observed at a 2θ angle of 31.72, 32.18, and 32.91, corresponding to the crystallographic planes (211), (112), and (300), respectively. The noticeable peaks and regions indicate the ordered crystalline structure of the material. These findings align with the HAP crystal structure [38]. On the other hand, ACP exhibits three peaks at 27.79, 31.03, and 34.39, which correspond to the crystallographic planes (214), (210), and (220), respectively. These peaks match with the primary peaks of an apatitic structure relating to β-tricalcium phosphate (β-TCP) [39]. 

The results support the extensively documented crystal structure of HAP, as evidenced by the characteristic peaks visible in its XRD patterns. In contrast, the XRD spectrum of ACP only showed a few low-intensity diffraction peaks, suggesting the presence of a minimal amount of crystalline material, which adopts a tricalcium beta-phosphate conformation. The significant absence of identifiable diffraction signals suggests that ACP is mostly composed of amorphous material.

### 3.2. FTIR-ATR of Electrospun Scaffolds

The infrared spectra shown in Figure 2, sections d–f show signals that match the functional groups found in PLA. Specifically, three higher intensity bands are visible at 1751 cm^−1^, 1184 cm^−1^, and 1087 cm^−1^, corresponding to the stretching of the C=O bond, the asymmetric stretching of the C-O-C (ester) group, and the asymmetric stretching of the C-O-C (ether) group, respectively [40].

The MHAP and MACP spectra (Figure 2e,f) exhibit vibrations that correspond to the amide I observed at 1651 cm^−1^ and amide II observed at 1546 cm^−1^ of collagen (Figure 2a) [41]. Furthermore, the vibrational modes of the PO_4_^−3^ group at 604 cm^−1^ and 567 cm^−1^, which are particular to calcium phosphates (HAP and ACP, Figure 2b,c), are also present [42]. However, due to the concentrations of these components in the membrane, the signals are weak and may overlap with the vibrations of the polyester.

### 3.3. XPS

The XPS spectra (Figure 2g–i) reveal the elemental composition and chemical bonding of the materials on the scaffold surface. The PLA scaffold (Figure 2g) exhibits characteristic C1s and O1s signals, indicating the presence of this aliphatic polyester. Similarly, the MHAP and MACP scaffolds (Figure 2h and 2i, respectively) display the C1s and O1s signals along with the N1s and Ca2p signals, indicating the existence of COL and calcium phosphate compounds on the scaffold surface. The deconvoluted signals from each XPS spectrum are depicted in Appendix A.

The C1s signals exhibit three distinct bands related to the C-C/C-H, C-O, and O-C=O bonds, ranging from 284.5 eV to 289.9 eV. This implies the existence of carbon in varying chemical environments, likely due to the different components of the scaffolds. The O1s signals reveal the O-C=O* and *O-C=O bands, ranging from 531.5 eV to 534.2 eV, indicating the presence of oxygen in distinct bonding environments, predominantly in carbonyl groups. The peak at 400.3 eV in the N1s signal suggests the presence of nitrogen in the collagen element of the scaffolds [43]. The calcium signal in MHAP exhibits two peaks, namely the 2p 1/2 and 2p 3/2 bonds, with binding energies of 347.7 eV and 351.3 eV, respectively [44]. These peaks indicate calcium’s binding to phosphate groups, which is in line with the literature. This confirms the presence of hydroxyapatite in the scaffold, as calcium is found in a chemical environment common to calcium phosphates [45,46].

A detection around 135 eV was expected for the phosphorus signal, but an abnormally low intensity was observed. This phenomenon is in line with Deng et al. results [44], which suggest that it could be due to the prevalent distribution of HAP particles within the fibers, without any exposure on their surface. Notably, no foreign elements are present in the materials used, as evidenced by the spectra.

### 3.4. SEM

The SEM images shown in Figure 3a–c display scaffolds featuring crosslinked fibers with even surfaces, uniform diameters, and no defects or beads. It is worth mentioning that the MHAP scaffold (Figure 3b) displays some agglomerations less than one micrometer in size, while the MACP scaffold (Figure 3c) exhibits agglomerations of approximately three micrometers deposited onto the electrospun fibers (green arrows). This phenomenon is caused by the non-homogeneous inorganic compound within the polymer solution.

To establish the fiber diameter, measurements were performed on every scaffold with ImageJ software (https://imagej.net/software/fiji/downloads (accessed on 1 May 2023)). The fiber diameter was determined by perpendicular measurements of their width at various points along their longitudinal axis. At least 60 fibers of each scaffold type were analyzed using ImageJ software (https://imagej.net/software/fiji/downloads (accessed on 27 July 2023)), from which the average diameter and standard deviation of the fibers constituting the electrospun scaffolds were calculated. The results demonstrated that the mean fiber diameter (Figure 3d–f) in the PLA scaffold was 412 ± 58 nm. In contrast, the mean diameters in the MHAP and MACP scaffolds were 240 ± 33 nm and 297 ± 81 nm, respectively. This indicates a reduction in the fiber diameter of up to 42% in the MHAP and MACP scaffolds due to the incorporation of COL and the calcium phosphate compounds into these scaffolds. Additionally, EDX (Figure 3g–i) also displays the elemental makeup of every scaffold, with MHAP and MACP retaining their Ca/P ratio.

### 3.5. DSC

Figure 4a–c illustrates the thermal behavior of three scaffolds. All scaffolds displayed two endothermic signals at 60 °C and 166 °C, as well as one exothermic signal at 87 °C. These signals were associated with the glass transition, melting, and crystallization of PLA, respectively [47]. Comparing the PLA scaffold (Figure 4a) to the MHAP and MACP scaffolds (Figure 4b,c), we observe a decrease in the crystallization temperature (Tc) and an increase in the melting temperature (Tm). The presence of COL and calcium phosphate compounds during the fiber formation in electrospinning modified the thermal behavior of the scaffold. Table 2 summarizes the temperatures and enthalpies calculated for each signal utilizing TA Instruments software (https://www.tainstruments.com/soporte/software-downloads-support/descargas/?lang=es (accessed on 27 July 2023)).

From the values of the crystallization and melting enthalpies of PLA in each scaffold, the percent crystallinity of the polymer in the scaffolds was determined using Equations (1) and (2) [48].
(1)$$Xc=\frac{\Delta H_{m}-\Delta H_{c}}{\Delta H_{m}^{*}\;}\times 100\%$$
where *ΔH_m_* is the enthalpy of fusion of the sample, *ΔH_c_* is the enthalpy of crystallization, and *ΔH*_m_* is the theoretical enthalpy of fusion of crystalline PLA (93.7 J/g) [49]. The absolute crystallinity of PLA in the scaffolds was calculated as *Xp*:(2)$$Xp=\frac{Xc}{w}\;$$
where *w* is the weight fraction of PLA in the scaffolds.

According to the calculations, the PLA, MHAP, and MACP scaffolds obtained crystallinity values of 26.38%, 31.47%, and 29.82%, respectively.

### 3.6. Young’s Modulus of Scaffolds

The uniaxial traction test of the scaffolding was carried out with a relative humidity of 55%. The Young’s modulus values (Figure 4d) and stress-strain diagram (Figure 4e,f) of each scaffold are presented in Figure 4.

The Young’s modulus values were calculated by determining the slope of the stress-strain diagram within the elastic zone for every scaffold. The PLA scaffold displayed values of 102 MPa ± 4 MPa, whereas the MHAP and MACP scaffolds showed a decrease with values of 34 MPa ± 7 MPa and 49 MPa ± 13 MPa, respectively.

The stress-strain diagram of electrospun PLA (Figure 4e) shows a deformation close to 7% with 4 MPa of uniaxial yield stress. Beyond the elastic limit, the polymer undergoes up to a 143% plastic deformation with an ultimate stress point of around 5.5 MPa. On the contrary, the elastic limit of both the MHAP and MACP scaffolds is reached (as shown in Figure 4f) at 1.45 MPa and 1.55 MPa, respectively, which corresponds to the yield stress, leading to a 5% and 4% deformation, respectively. Additionally, the plastic deformation is reduced to 20% for MHAP and 32% for MACP in these electrospun polymer matrices, each with an ultimate stress point of approximately 1.9 MPa and 2.4 MPa, respectively. This suggests that MHPA and MACP have lower elastic properties (Young’s modulus and yield stress) compared to PLA, resulting in less mechanical resistance. Additionally, PLA has greater ductility when plastically deformed, while MHPA and MACP are more brittle.

### 3.7. Adhesion and Proliferation Cellular

Fluorescence microscopy was used to observe the distribution and adhesion of hWJ-MSCs-GFP on electrospun scaffolds. Cells with an elongated morphology and extended filopodia were present on all three scaffolds from the first day of culture. However, the adhesion of cells on the MACP scaffold (Figure 5g–i) was greater than that on the MHAP (Figure 5d–f) and PLA (Figure 5a–c) scaffolds, with more cells present on the scaffold on days 3 and 5 compared to the other two scaffolds. These results indicate that the MACP scaffold provides better support for cell adhesion and proliferation.

Figure 5j–l shows the confocal microscopy images of the cytoskeleton (red, phalloidin) and nuclei (blue, DAPI) of hWJ-MSCs on the electrospun scaffolds. Several spindle-shaped spreading cells adhered to the scaffolds. Furthermore, all three scaffold types show a well-developed cytoskeletal arrangement characterized by network-like structures and robust interconnections with neighboring cells. Correspondingly, the fluorescence analysis shows that the incorporation of COL and calcium phosphate compounds increases cell density. In particular, the MACP scaffold (Figure 5l) shows the highest cell density among the scaffolds.

The effects of the scaffolds on cell viability were examined by resazurin assay (Figure 6a). The non-cytotoxicity and cell compatibility related to the MACP and MHAP scaffolds were observed compared to the control (hWJ-MSCs on TCP) at 24 and 48 h, which confirms the biocompatible nature and safety of these scaffolds for their possible use in clinical applications [50]. 

Similarly, the proliferation of hWJ-MSCs seeded on the PLA, MHAP, MACP and TCP scaffolds is shown in Figure 6b. On the first day of the culture, cell proliferation was 40% in the PLA scaffolds, whereas it exceeded 50% in the MHAP and MACP scaffolds. Subsequently, the cell proliferation in the MACP and MHAP scaffolds surpassed 60% and 75% after three and five days, respectively, while the PLA scaffold attained values approximating 50%. Note that starting from the third day of culture, the MHAP and MACP scaffolds displayed a significant deviation compared to the PLA scaffold, which suggests the impact of COL and CaPs on cell proliferation within these scaffolds. Additionally, significant differences in cell proliferation were observed between the scaffolds and the control (TCP) at all evaluated time points.

### 3.8. Quantification of Growth Factors

The growth factors quantified in this study play a role in bone healing [51]. We measured the secretion of basic fibroblast growth factor (bFGF), transforming growth factor beta 1 (TGF-β1), angiopoietin-1, and platelet-derived growth factor bb (PDGF-bb) in all conditions (PLA, MACP, and MHAP).

Additionally, we analyzed the culture-supplemented medium to determine the concentration of growth factors in the medium. Likewise, the cells cultured in standard tissue culture plates (TCP) were used as a control to determine the secretory factors under typical culture conditions. The secretion of bFGF by cells cultured in the MACP scaffold was the highest after 24 and 48 h of culture. At 48 h, the MACP constructs had a bFGF concentration of 133 pg/mL, compared to 93 pg/mL in the MHAP, 30 pg/mL in the polylactic acid (PLA) scaffold, and 8.6 pg/mL in the TCP. These concentrations exhibited significant differences (*p* < 0.05 for PLA and *p* < 0.01 for TCP) (Figure 7a).

On the other hand, both TGF-β1 and angiopoietin-1 demonstrated similar behavior in all conditions. Their values were lower (7688 pg/mL and 4190 pg/mL, respectively) than those found in the culture medium for up to 48 h. However, after 72 h, their production became evident, with TGF-β1 reaching 10,300 pg/mL and angiopoietin-1 reaching 12,900 pg/mL (see Figure 7b,c). PDGF-bb, on the other hand, showed a trend of consumption in all treatments, with an initial concentration of 1900 pg/mL in the culture medium. It was gradually reduced in all treatments to values ranging from 120 pg/mL (PLA) to 15 pg/mL (TCP) over the evaluated period (Figure 7d).

### 3.9. Calcium Phosphate Compounds-Containing Scaffolds Induce the Osteogenic Differentiation of hWJ-MSCs

Considering the potential of hWJ-MSCs to differentiate toward osteogenic lineages, we finally sought to test whether hWJ-MSCs cultured on PLA, MHAP, and MACP could induce their differentiation toward osteogenic lineages. The hWJ-MSCs were seeded on the PLA, MHAP, and MACP scaffolds and cultured with the supplemented DMEM and osteogenic differentiation medium. After 7 days of culture, samples of PLA/hWJ-MSCs, MHAP/hWJ-MSCs, and MACP/hWJ-MSCs were fixed for evaluation by SEM, alizarin red staining, and immunohistochemical detection of osteocalcin.

SEM images showed cell adhesion in all scaffolds, but the inherently hydrophobic nature of the PLA scaffold resulted in a lower number of cells present (Figure 8a–l). Conversely, the MHAP scaffold showed higher cell density when cultured with the osteogenic differentiation medium. Furthermore, when hWJ-MSCs were seeded onto the MACP scaffold, they formed a uniform and dense cell layer covering the electrospun fibers, with cells evenly distributed throughout the scaffold. Extracellular matrix (ECM) formation and prominent cell extensions were observed, indicating the presence of a three-dimensional network of proteins and molecules secreted by the cells into their environment. This network provides structural and biochemical support and suggests a beneficial interaction between cells and the scaffold, enabling cell adhesion and growth in the scaffold environment [52].

To evaluate the mineralization of hWJ-MSCs on the scaffolds cultured with supplemented DMEM and osteogenic culture media, we performed alizarin red staining after 7 days of culture. The staining shows an increase in staining intensity for both the MHAP and MACP scaffolds (Figure 8m). This increase in staining is evident in the scaffolds cultured in both supplemented DMEM and osteogenic differentiation media. The formation of calcium-rich deposits in the scaffold indicates the ability of these matrices to promote bone differentiation processes [53,54]. In the PLA/hWJ-MSCs, low formation of calcium deposits was observed in the scaffold exposed to the osteogenic differentiation medium [54]. No mineralization was observed in the cells seeded in TCP (Figure 8m).

Similarly, immunohistochemical analyses performed on the MHAP (Figure 9a,b,e,f) and MHAP (Figure 9c,d,g,h) scaffolds show staining for osteocalcin, indicating the potential of these scaffolds to induce osteogenic differentiation. The PLA and TCP controls (Appendix A) did not show staining, which may indicate that CaPs in the scaffolds induce cell differentiation to the osteogenic lineage. More intense staining for osteocalcin expression was observed in the MACP scaffold, suggesting a microenvironment conducive to the process of osteogenesis. Similarly, the MHAP scaffold showed a slightly reduced but still notable level of osteocalcin presence, indicating its potential to stimulate osteogenic activity. These results highlight the influence of scaffold composition on cellular behavior and suggest that both the MACP and MHAP scaffolds have great potential in bone tissue engineering.

## 4. Discussion

A tissue engineering strategy that relies on scaffolds seeded with MSCs is a promising approach for treating bone defects. Currently, we have a good understanding of how various scaffold characteristics can affect cell behavior and the efficiency of the regenerative process. For instance, it is known how the scaffold materials can affect cell infiltration, viability, migration, and differentiation. HAP is one of the main inorganic compounds used in bone tissue engineering matrices. However, studies have shown a slow dissolution rate that hinders bone repair [55]. As a result, researchers have turned to ACP as a substitute. ACP has a structural unit similar to HAP, which allows it to play a critical role in apatite formation both in vitro and in vivo. ACP has a disordered structure that confers high reactivity in body fluids, resulting in high solubility and the rapid precipitation of apatite [56].

SEM images demonstrated that ACP exhibits a rougher surface and greater dimensional variability compared to HAP, which had a smoother surface and uniform sizes. Also, the dimensions of the hydroxyapatites affect their distribution in the scaffolds, since the size of the HAP allows it to be more easily and homogeneously distributed on the electrospun fibers. In contrast, the MACP scaffold deposits amorphous calcium phosphate in agglomerations of mostly over 1 micron inside the scaffold and a small amount on the electrospun fibers.

Regarding the EDX spectrum, it indicates that ACP has a Ca/P ratio indicative of a predominantly amorphous ceramic, whereas HAP has a ratio indicative of a crystalline ceramic [57]. When comparing our amorphous calcium phosphate with the results of Abidi et al. [30], differences in Ca/P were found. These differences are attributed to variations in the pH level of the solution during the bioceramic synthesis. The XRD analysis allowed the identification of the ACP phase, characterized by its minimal crystallinity, with no other phases detected within the defined limits of detection. The data indicates that the detected signal aligns with the peaks of a β-TCP structure. However, it should be noted that the crystalline structural formation of this bioceramic necessitates temperatures exceeding 500 °C [58]. Considering the synthesis conditions, morphology, Ca/P ratio, and XRD results of the ACP ceramic, it can be inferred that it is largely composed of amorphous CaP. The compositional profile of ACP is particularly dependent on the pH level of the solution and the concentrations of the mixed solutions, resulting in different Ca/P ratios ranging from 1.18 to 2.5 [59]. There is an ongoing debate regarding the apatitic nature of ACP, with one perspective suggesting that its apatitic structure consists of extremely fine crystals, rendering it X-ray amorphous. Alternatively, one hypothesis suggests that the basic structural unit of ACP is a roughly spherical ion cluster with a diameter of 9.5 Å and a composition of Ca_9_(PO_4_)_6_ [60]. These clusters are the initial nuclei during the crystallization of hydroxyapatite, supporting a model that elucidates hydroxyapatite formation as a stepwise assembly of these basic units [61].

On the other hand, the morphological evaluation of electrospun scaffolds revealed that the fibers obtained from all treatments had a random submicron distribution without defects or beads. This confirms that the parameters used were appropriate for obtaining smooth and homogeneous electrospun fibers. Furthermore, imaging revealed a decrease in fiber diameter in the MACP and MHAP scaffolds due to COL and CaPs compounds. These materials increase polarity and Coulomb interaction forces, thereby increasing electrostatic repulsion, reducing fiber diameter, and producing high surface area scaffolds [62]. This feature helps to create a network of interconnected micropores that promote cell signaling and the transport of nutrients and factors during tissue regeneration [63].

Another important feature of a bone scaffold is its chemical composition, which must have functional groups, such as those provided by COL and HAP, that promote cell adhesion, proliferation, and differentiation. The incorporation of COL and CaPs compounds into the scaffolds was confirmed by the visualization of amide groups I and II and phosphate groups by FTIR-ATR and N1s and Ca2p signals by XPS. In addition, the DSC analysis has demonstrated that these materials enhance the melting enthalpy of scaffolds, which is linked to the recrystallization of the polyester. Because PLA crystallizes at a slow rate and its glass transition temperature (Tg) is greater than the ambient temperature (T amb), a crystalline fraction of this polyester exists that facilitates recrystallization during electrospinning [64]. The presence of hydroxyapatites constrains the mobility of the PLA molecular chain during recrystallization. It serves as a site for nucleation that promotes the formation of flawed crystals [65], which demand less energy but a higher temperature to liquefy, leading to a Tm shift in these frameworks [66]. The thermal changes of the scaffolds (Tg, Tc, and Tm) are located at temperatures exceeding 38 °C; thus, the physical temperature would not impact the features of the PLA, MHAP, and MACP frameworks. In addition, polymer degradation may be facilitated by microstructural changes, as highly crystalline PLA takes longer to degrade [67]. Also, the polymer microstructure underwent changes due to the aggregation of organic and inorganic compounds. The addition of CaPs to the MHAP and MACP scaffolds significantly affected their mechanical properties. This was due to a decrease in the yield stress and Young’s modulus in the scaffolds, mainly caused by the aggregation of CaPs. Such an aggregation affected the intermolecular linkages within the PLA molecules, resulting in a higher incidence of mesostructural defects within the polymer. As a result, the ability of the scaffolds to stretch before reaching the failure point was greatly reduced [25,36,68].

The in vitro cellular response of the electrospun scaffolds was evaluated with hWJ-MSCs. The scaffolds demonstrated non-cytotoxicity by meeting the ISO 10993-5 standards with values above 75%, which is similar to other literature reports using these materials [69,70]. The MACP scaffold demonstrated greater in vitro adhesion and proliferation of hWJ-MSCs than both the MHAP and PLA scaffolds. These findings align with those presented in Zhao et al.’s study, which utilized electrospun ACP scaffolds and MG63 cells [25]. Moreover, cellular receptors, including glycoprotein VI, integrins, discoidin domain receptors, and receptors with an affinity for non-collagenous domains, detect specific COL peptide sequences that regulate cell behaviors, such as adhesion, differentiation, and proliferation [71]. COL also plays a crucial role in tissue regeneration by contributing to the retention, local storage, and delivery of growth factors or cytokines [72].

Additionally, the Luminex analysis revealed that cells seeded on the MACP scaffolds during the initial 48 h secreted higher levels of bFGF compared to MHAP, PLA, and the control. bFGF is a critical growth factor for bone regeneration and its presence influences many mesodermal and neuroectodermal cell types. As for TGF-β1 and angiopoietin 1, the production of these growth factors was observed after 48 h of culture. The concentrations at each time point were comparable between the scaffolds, except for a higher TGF-β1 production observed in the PLA scaffold at 72 h. However, by 96 h, the levels of this factor in the PLA scaffold were similar to those in the other treatments. Finally, the concentration of PDGFbb decreases in all treatments over time, suggesting that hWJ-MSCs are consuming this factor. This phenomenon is attributed to the high concentration of this factor in the platelet lysate [73]; therefore, the cells do not produce it but rather consume it. 

TGF-β1 and PDGF-bb factors are stimulated upon injury to promote the proliferation of mesenchymal cells and osteoblasts in fractures and experimental bone defects (37). Additionally, Angiopoietin-1 facilitates vascular growth and promotes the formation of new blood vessels to enable oxygen, nutrients, osteoinductive factors, and stem cells to reach the damaged area. It also aids in the formation of the vascular wall, promoting the migration of osteoblast precursor cells to the target site [74]. The secretion of these growth factors stimulates the recruitment and storage of growth factors involved in bone formation and repair [75].

Notably, the MACP scaffolds not only facilitate cell adhesion and proliferation and stimulate the release of growth factors associated with bone repair, but also have the remarkable ability to induce the formation of calcium reservoirs, an extracellular matrix, and essential bone regeneration peptides, including osteocalcin by hWJ-MSCs. These findings provide further insight into the bioactive potential of the MACP scaffolds in the field of bone regeneration. The extracellular matrix is actively synthesized by the cells on the scaffolds, either in the supplemented DMEM medium or in the specialized osteogenic differentiation medium, to the extent that it completely covers the scaffold surface. The observed extracellular matrix indicates a beneficial interaction between the scaffold and the cells. This interaction suggests that the environment is conducive to bone differentiation and repair [76]. With respect to osteocalcin, the staining intensity indicates increased production of this protein in the MACP scaffolds with hWJ-MSCs. OCN, a major non-collagenous protein in the bone matrix, is present at high levels during bone formation and has the ability to directly or indirectly influence bone mass, mineral size, and alignment [77]. 

## 5. Conclusions

This study comprehensively characterizes hydroxyapatite (HAP) and amorphous calcium phosphate (ACP) and their incorporation into electrospun scaffolds, highlighting their potential for bone tissue engineering. The SEM analysis reveals distinct morphological differences between HAP and ACP, with HAP exhibiting uniform particle size and a smooth surface, while ACP forms irregular aggregates with a rough texture. The Ca/P ratio variations are verified by the EDX analysis. Successful scaffold incorporation is demonstrated by the FTIR-ATR spectra, and calcium binding to phosphate groups is revealed by the XPS analysis. The addition of collagen and calcium phosphate causes changes in thermal behavior, as indicated by the DSC results. Young’s modulus values evidence the changes in scaffold mechanics. Cellular studies showed that the MACP scaffolds significantly increased the adhesion, proliferation, osteogenic differentiation, and growth factor secretion of hWJ-MSCs. In addition, SEM and alizarin red staining demonstrated their mineralization potential, positioning the MACP scaffolds as promising candidates for bone tissue engineering. These results highlight the critical role of scaffold composition in influencing cellular behavior and tissue regeneration. Future research aims to evaluate the mineralization of MACP in simulated body fluid (SBF) following ISO 23317:2014 standards to demonstrate the viability of calcium phosphates incorporated in scaffold matrices for use in biomedical applications. It should be noted that the degradation of these types of scaffolds has yet to be comprehensively understood. Therefore, our ongoing research endeavors to address these critical knowledge gaps. This study offers insight into the intricate function of calcium phosphates in regulating mineralization processes and degradation mechanisms of scaffolds, thereby improving their potential for biomedical applications.

## Figures and Tables

**Figure 1 pharmaceutics-15-02529-f001:**
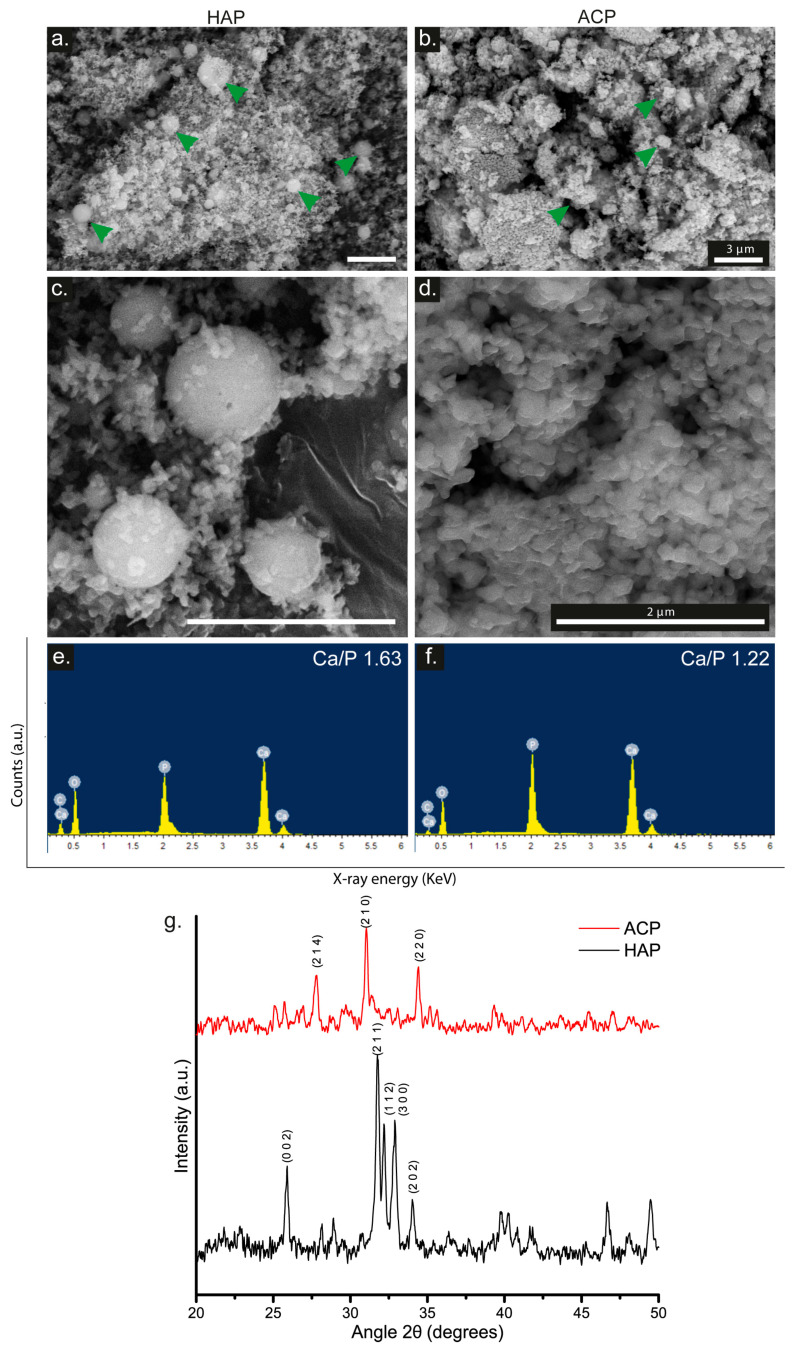
SEM images of (**a**,**c**) HAP and (**b**,**d**) ACP (Scale Bar = 3 µm; green arrows indicate the bioceramic). EDS spectrum of (**e**) HAP and (**f**) ACP. (**g**) XRD pattern of HAP and ACP.

**Figure 2 pharmaceutics-15-02529-f002:**
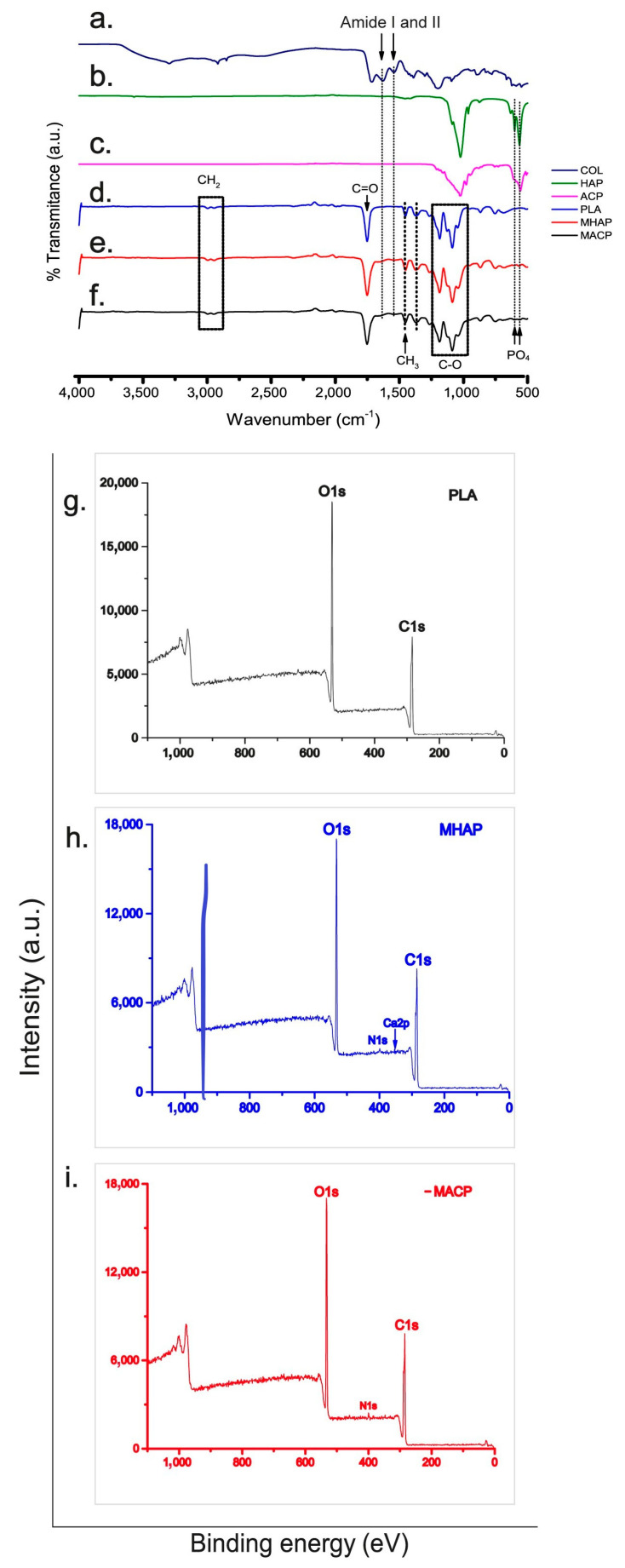
Infrared spectra of (**a**) COL, (**b**) HAP, (**c**) ACP, (**d**) PLA, (**e**) MHAP, and (**f**) MACP. XPS spectrum of (**g**) PLA, (**h**) MHAP, and (**i**) MACP scaffolds.

**Figure 3 pharmaceutics-15-02529-f003:**
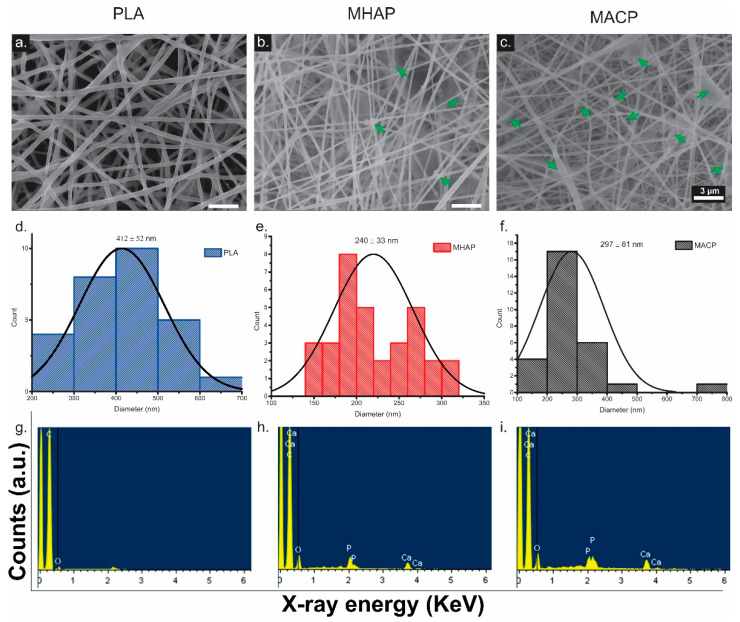
SEM images of electrospun scaffolds of (**a**) PLA, (**b**) MHAP, and (**c**) MACP. (Scale bar = 3 µm, green arrows indicate CaP). Histogram of fiber diameters of (**d**) PLA, (**e**) MHAP, and (**f**) MACP electrospun scaffolds. EDX spectra of (**g**) PLA, (**h**) MHAP, and (**i**) MACP scaffolds.

**Figure 4 pharmaceutics-15-02529-f004:**
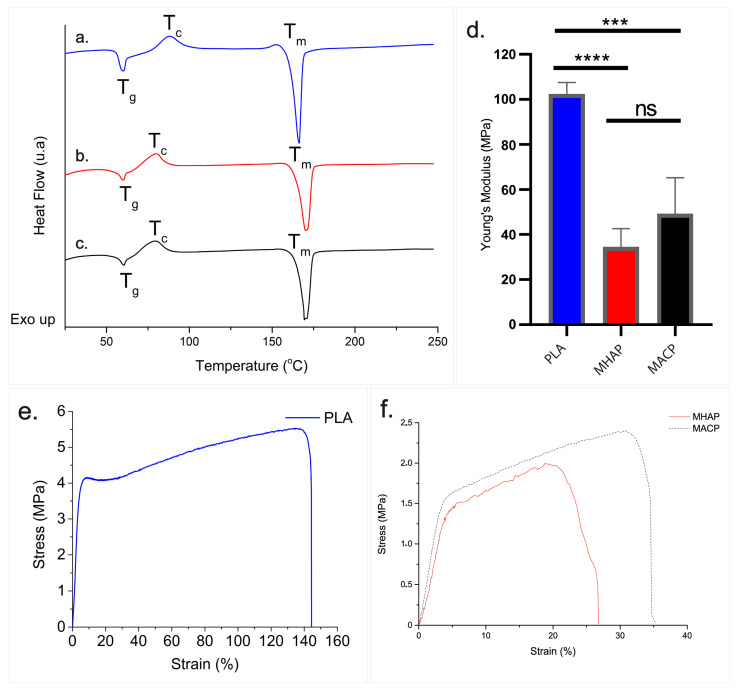
DSC of (**a**) PLA, (**b**) MHAP, and (**c**) MACP electrospun scaffolds. (**d**) Young’s modulus of the scaffolds (*** *p* < 0.001, **** *p* < 0.0001, ns: no significative). Stress vs. strain plot of (**e**) PLA and (**f**) MHAP and MACP electrospun scaffolds.

**Figure 5 pharmaceutics-15-02529-f005:**
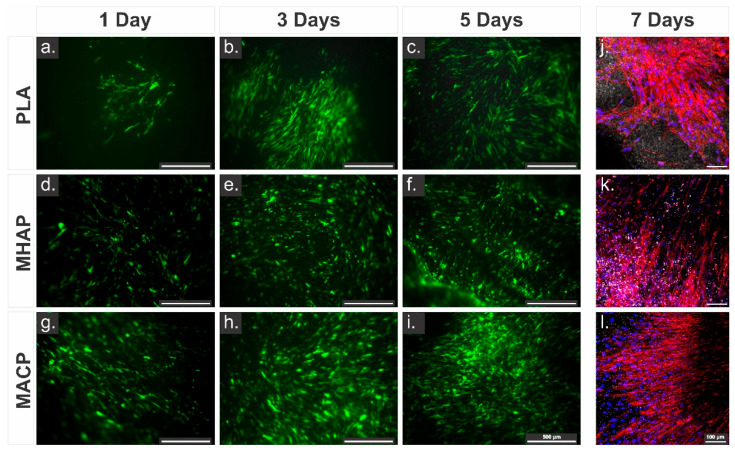
hWJ-MSCs-GFP seeded on electrospun scaffolds and imaged at different time points. (**a**–**c**) PLA scaffolds, (**d**–**f**) MHAP scaffolds, and (**g**–**i**) MACP scaffolds (Scale bar = 500 µm, Green color show the presence of GFP-transduced cells.) and confocal images of hWJ-MSCs staining with DAPI (blue) and red phalloidin (red) seeded 7 days on (**j**) PLA, (**k**) MHAP, and (**l**) MACP scaffolds (Scale bar = 100 µm).

**Figure 6 pharmaceutics-15-02529-f006:**
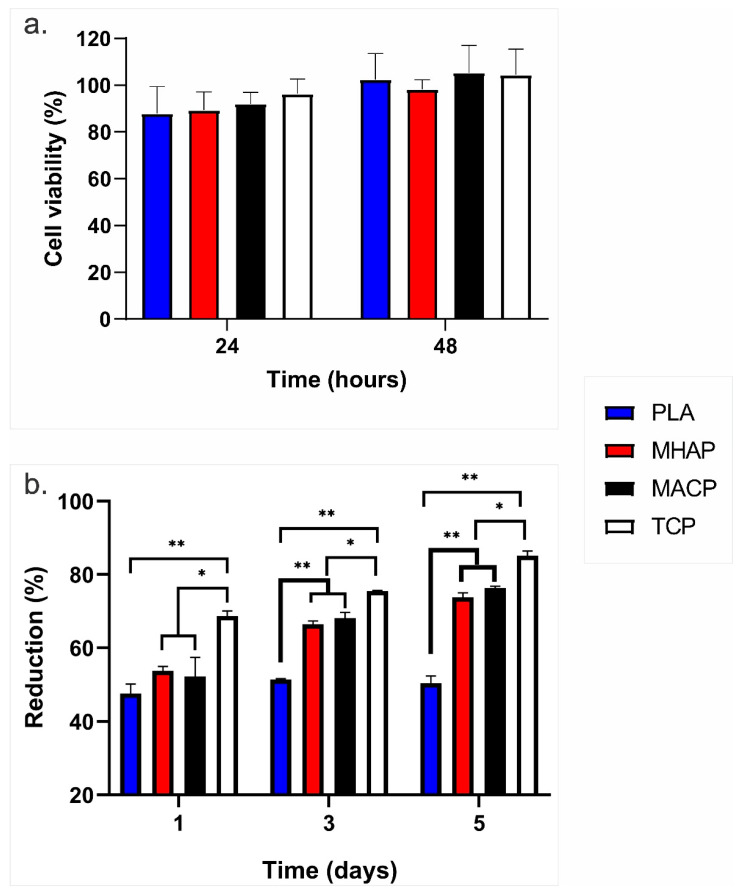
(**a**) Cell viability of hWJ-MSCs cultured in direct contact with PLA, MHAP, and MACP scaffolds for 24 and 48 h. (**b**) Proliferation of hWJ-MSCs seeded on PLA, MHAP, MACP, and TCP scaffolds (* *p* < 0.05, ** *p* < 0.01).

**Figure 7 pharmaceutics-15-02529-f007:**
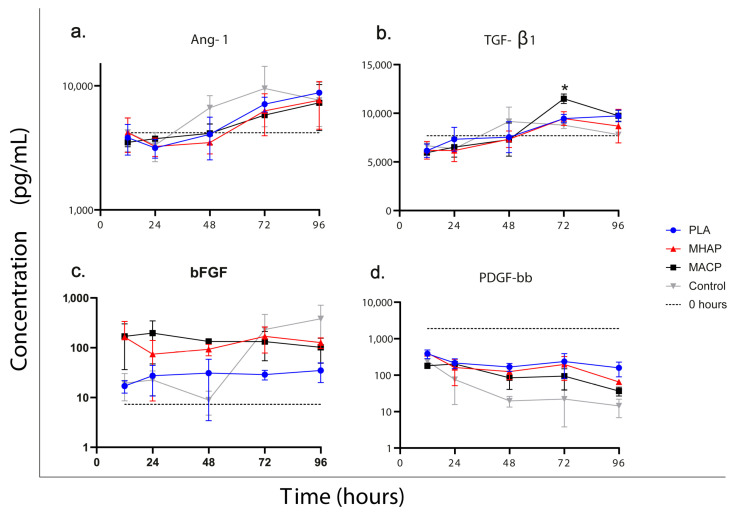
Quantification of (**a**) angiopoietin-1, (**b**) transforming growth factor beta 1, (**c**) basic fibroblast growth factor and (**d**) platelet-derived growth factor bb evaluated in hWJ-MSCs seeded on PLA, MHAP, MACP, and TCP scaffolds. Different signs in the same graph indicate significant differences (* *p* < 0.05) PLA vs. other treatments.

**Figure 8 pharmaceutics-15-02529-f008:**
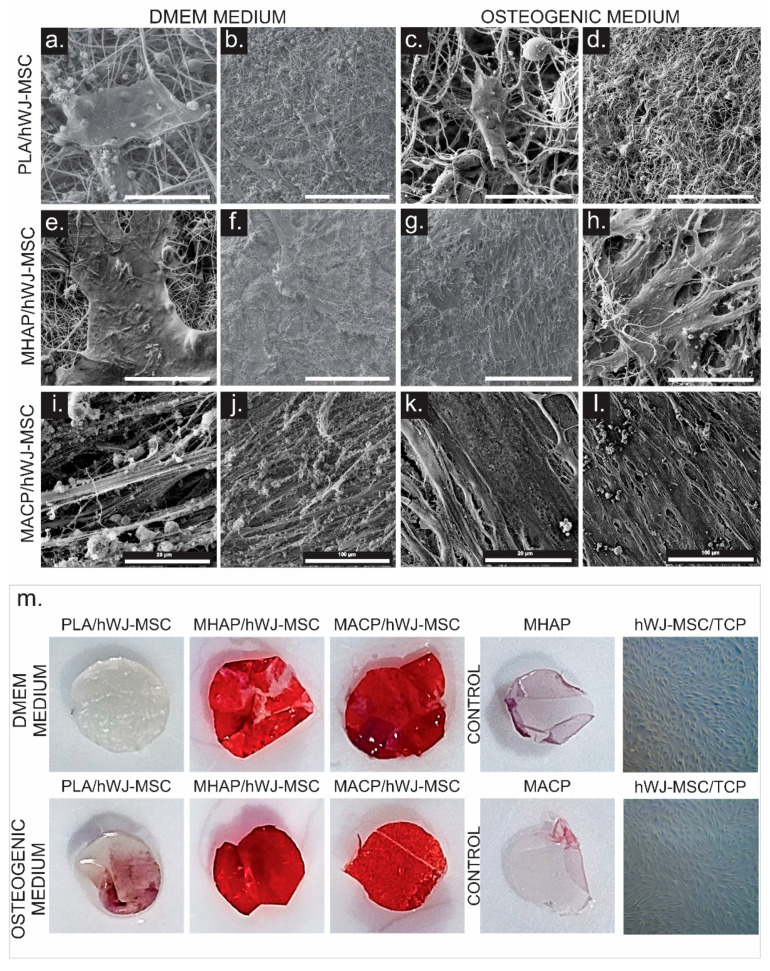
SEM images of hWJ-MSCs seeded on PLA, MHAP, and MACP scaffolds. Images (**a**,**b**,**e**,**f**,**i**,**j**) show cells cultured with the supplemented DMEM medium, while images (**c**,**d**,**g**,**h**,**k**,**l**) show cells cultured with the osteogenic differentiation medium. Image (**m**) shows alizarin red staining performed on PLA, MHAP, and MACP scaffolds with hWJ-MSCs seeded with both the supplemented DMEM medium and osteogenic differentiation medium.

**Figure 9 pharmaceutics-15-02529-f009:**
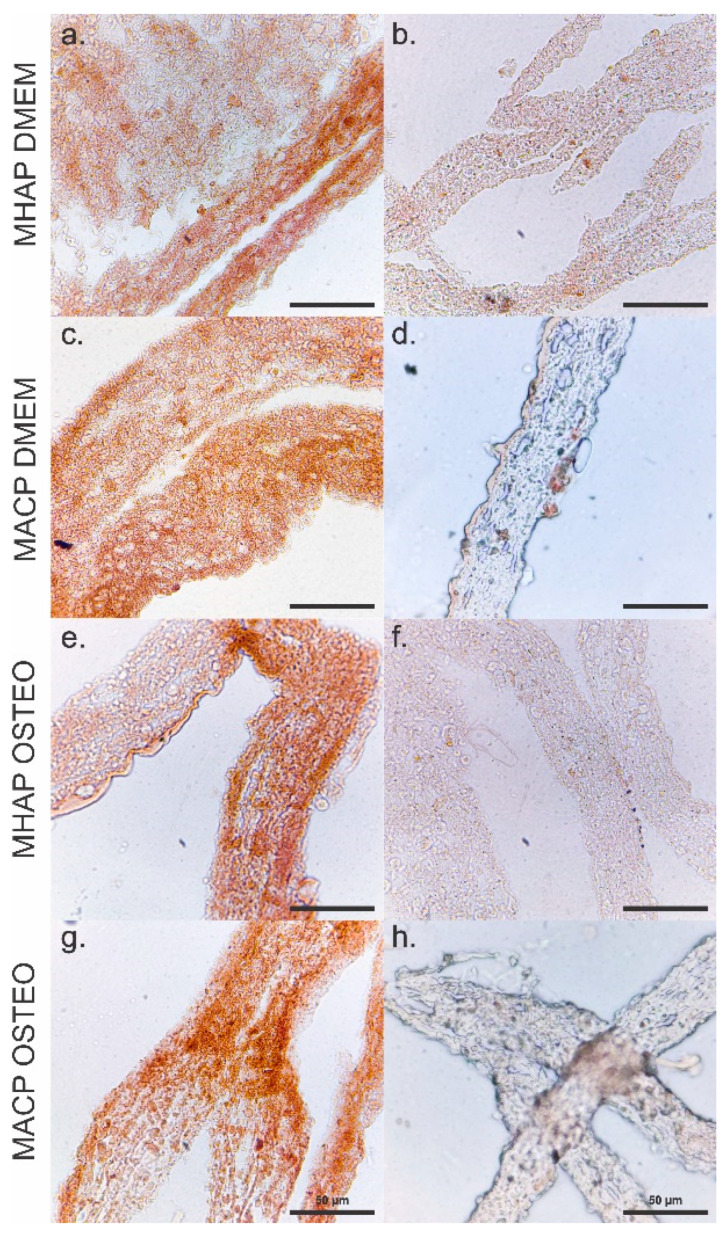
Osteocalcin immunohistochemical staining of hWJ-MSCs seeded on MHAP and MACP scaffolds after 7 days of culture with (**a**,**c**) and without (**e**,**g**) osteogenic media condition. Negative control staining for scaffolds cultured in the DMEM medium (**b**,**d**) and scaffolds cultured in the osteogenic differentiation medium (**f**,**h**). Scale bar = 50 µm.

**Table 1 pharmaceutics-15-02529-t001:** Specifications of solutions prepared for electrospinning.

Nomenclature	Calcium Phosphate	PLA (mg/mL)	COL (mg/mL)	CaP (mg/mL)
MHAP	Hydroxyapatite	100	10	10
MACP	Amorphous	100	10	10
PLA	N/A	100	-	-

**Table 2 pharmaceutics-15-02529-t002:** Temperatures and enthalpies of MHAP, MACP, and PLA scaffolds.

Scaffold	Tg (°C)	*ΔHg* (J/°C)	Tc (°C)	*ΔHc* (J/°C)	Tm (°C)	*ΔHm* (J/°C)
MHAP	60.29	2.62	79.88	10.80	170.16	35.28
MACP	60.59	4.14	78.64	11.76	169.42	35.04
PLA	59.47	6.91	87.76	13.79	166.16	38.51

## Data Availability

The datasets used and/or analyzed during the current study are available from the corresponding author.

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
