# Peer review of "Electrospun Scaffolds of Polylactic Acid, Collagen, and Amorphous Calcium Phosphate for Bone Repair"

_pharmaceutics, 2023, doi:10.3390/pharmaceutics15112529_

Round 1

Reviewer 1 Report

Dear Authors, 
please see the attached file.

The manuscript is understandable in terms of English language.

Author Response

In the manuscript titled “Electrospun Scaffolds of Polylactic Acid, Collagen, and Amorphous Calcium Phosphate for Bone Repair” by William Cárdenas-Aguazaco et al., the Authors proposed a novel composite-type matrix in the form of scaffolds that are based on PLA with osteogenic additives – collagen and calcium phosphates. The rationale of the work is well explained with introduced elements of novelty. The work includes the necessary experiments and provides the proper description of the obtained results. Nonetheless, additional aspects should be considered as this article requires major revision, otherwise, it should not be published in the presented form.

  1. Based on the presented results, in my opinion, the Authors did not confirm the advantage of using ACP vs. HAP in the scaffolds explicitly. The mineralization studies in simulated body fluid (SBF) should be carried out following ISO 23317:2014 procedure, to decide which type of calcium phosphate is more suitable. Did the Authors compare the mineralization properties of final scaffolds in SBF or, at least, the mineralization properties of ACP and HAP? Such a study would give the Authors both the arguments for using specific calcium phosphate in the scaffold and valuable conclusions. The study on the degradation of each type of scaffold might be also useful. 

Thank you for this suggestion. The study is a first approach in comparing the properties of electrospun scaffolds containing amorphous calcium phosphate (ACP) or hydroxyapatite (HAP) at both the physicochemical and biological level. Our aim was to analyze the physicochemical properties and cellular response to the scaffolds. We did not perform specific mineralization tests in simulated body fluid (SBF) according to ISO 23317:2014. Nevertheless, we recognize the importance of such tests and consider them crucial for future studies to assess the suitability of the calcium phosphates in our scaffolds. Similarly, it is important to note that we did not perform a detailed comparison of the mineralization properties between ACP and HAP, as this was not a primary objective of this study. However, we acknowledge that such an analysis may provide further evidence to support the selection of a precise calcium phosphate in our scaffolds and enhance our results. In this regard, it is important to note that conclusive results on scaffold degradation have not been obtained. Our ongoing research aims to address these critical issues and to elucidate how calcium phosphates influence the mineralization and degradation of scaffolds used in biomedical applications.

  1. The Authors obtained the calcium phosphate following the procedure proposed by Abidi et Murtaza J. Mater. Sci. Technol. (30) 2014. In the cited manuscript, it is written that “HA powder prepared in this study is a mixture of crystalline as well as amorphous phases with the major content being the amorphous phase of HA”. Thus, the Authors should either prove that their calcium phosphate is completely amorphous (FTIR and XRD analyses must be added) or change their statement. The Ca/P ratio of the obtained calcium phosphate at 100°C (Ca/P=1.22) differs from that in the referential manuscript (Ca/P~1.5). Please comment this result. 

We extend our gratitude to the esteemed reviewer for their comments and recommendations pertaining to the origin and characterization of the calcium phosphate material used in our study. We acknowledge the importance of clarifying the nature of the calcium phosphate and ensuring the consistency of our findings with the reference material. The discrepancy in the Ca/P ratio observed at 100°C compared to the reference manuscript (Ca/P~1.5) has been duly noted:

“Regarding the EDX spectrum, it indicates that ACP has a Ca/P ratio indicative of a predominantly amorphous ceramic, whereas HAP has a ratio indicative of a crystalline ceramic [54]. When comparing our amorphous calcium phosphate with the results of Abidi et al. [30], differences in Ca/P were found. These differences are attributed to variations in the pH of the solution during bioceramic synthesis. It has been shown that this factor is fundamental in calcium phosphate synthesis processes, with values of pH less than 9 resulting in mostly amorphous calcium phosphates [55].”  We make note of this in our Discussion section (line 509).

  1. More representative SEM micrographs of HAP and obtained CAP should be provided to confirm the differences in morphology explicitly. What are the white arrows on the micrographs? The Ca/P ratio should be also presented in the EDX spectra. 

Your feedback has been valuable in improving our work. Regarding your specific points:

**SEM Micrographs**: More representative SEM micrographs of HAP and Calcium Amorphous Phosphate (obtained ACP) could enhance the clarity of our findings. We have provided SEM images that better illustrate the morphology differences between these materials.

**White Arrows**:  We regret any confusion caused by the white arrows in the micrographs. The arrows were used to aid in observing bioceramics. We changed the color to green to improve its visualization.

We have provided the atomic percent of calcium (Ca) and phosphorus (P) in the following paragraph.

“EDX analysis evaluated the elemental composition of HAP and ACP, as shown in Figure 1c and Figure 1d, respectively. The EDX spectra show the presence of calcium (Ca) and phosphorus (P) signals in both materials, confirming their identical elemental composition. Specifically, HAP shows atomic percentages of 3.37% for P and 5.50% for Ca, while ACP shows higher atomic percentages of 6.74% for P and 8.23% for Ca. However, the differences in the Ca/P ratio between the two compounds (1.63 for HAP and 1.22 for ACP) suggest variations in their degree of crystallinity. Based on the data presented by Munir et al. [35], it is clear that HAP is equivalent to calcium hydroxyapatite, while ACP is classified as amorphous calcium phosphate.”  We make note of this in our HAP and ACP characterization section (line 253).

  1. In the case of FTIR-ATR, the additional spectrum of collagen should be provided. Moreover, I recommend adding and comparing the FTIR-ATR spectra of HAP and CAP, i.e. in Fig. 1.  

In accordance with the reviewer's comment, we have added the FTIR-ATR spectrum for collagen, hydroxyapatite (HAP), and the obtained calcium phosphate (ACP). This addition has led to a more extensive characterization of the materials utilized in our study. The new results can be found in Figure 2a-2c.

  1. The Authors stated that observed agglomerates in scaffolds’ structure are either HAP or ACP particles. Please mark these agglomerates in SEM micrographs and provide EDX spectra of filament and agglomerate. Moreover, the inserted in Fig. 3 histograms are of low quality and hard to interpret. Please provide a better description of how the diameters of the filaments were measured. 

We thank the reviewer for these comments. The observed agglomerates in the SEM micrographs have been marked with green arrows and accompanied by corresponding energy dispersive X-ray spectroscopy (EDX) spectra of the fibers and agglomerates. This information enhances the characterization of these scaffolds.

We regret any difficulty in interpreting the quality of the histograms shown in Figure 3. We have improved and refined the quality of these histograms to ensure that they provide a clear and accurate representation of fiber diameters. In addition, we have provided a comprehensive description of the methodology used to measure these diameters, thereby increasing the transparency of our measurement process:

"Fiber diameter was determined by perpendicular measurements of their width at various points along their longitudinal axis. At least 60 fibers of each scaffold type were analyzed using ImageJ software, from which the average diameter and standard deviation of the fibers constituting the electrospun scaffolds were calculated." We make note of this in our SEM section (line 310).

  1. Deconvoluted peaks of XPS spectra for each element should be provided, otherwise, it is hard to interpret possible interactions inside the matrix. Why phosphorus peak is not present in XPS spectra? In the references 29 and 30, I could not find the description of Ca-P interactions. Please explain this in more detail. 

We agree that presenting deconvoluted peaks (Supplementary Figure 1) for each element allows for a more comprehensive and informative assessment of potential interactions within the matrix. Our revised manuscript includes deconvoluted peaks for carbon, oxygen, calcium and nitrogen, allowing for a more detailed examination of chemical states and interactions.

Regarding the absence of the phosphorus peak in the XPS spectra, we have included the following observation: A detection around 135 eV was expected for the phosphorus signal, but an abnormally low intensity was observed. This phenomenon is in agreement with the results of Deng et al. (36), which suggest that it could be due to the widespread distribution of HAP particles inside the fibers, without any exposure on their surface. We make note of this in our XPS section (line 295).

Similarly, we have replaced references 29 and 30 with others that provide a more thorough explanation of Ca-P interactions, thus improving the reader's understanding.

  1. The Authors stated that polymeric solutions of PLA/COL/HAP (MHAP) or PLA/COL/ACP (MACP) were prepared according to the ratios proposed by Abidi et al [20]. What are these ratios? Moreover, it is written that materials were dissolved in trifluoroethanol. I would rather use “suspended” due to the presence of calcium phosphates in the formulation. In Table 1, are the concentrations of each material given in mg per mL of trifluoroethanol? Did the Authors measure the residual level of trifluoroethanol after the electrospinning?  

We apologize for our lack of clarity. We have now revised as follows:

We have changed the text in the citation and reference to: MHAP and MACP solutions were mixed at a weight ratio of 10:1:1 (PLA:Ca/P) according to the ratios proposed by Akkouch et al [28], with some modifications (see Table 1). We will include a clear description of these ratios in the revised manuscript in response to your request. We make note of this in our Electrospun scaffold fabrication section (line 123).

Concerns were raised about the use of "dissolved" instead of "suspended" in the context of trifluoroethanol due to the presence of calcium phosphates. To accurately reflect the formulation process, we have made the necessary change to indicate that the materials were "suspended" in trifluoroethanol.

In addition, the concentrations of each material in Table 1 are given in mg per mL of trifluoroethanol, which represents the solute-to-solvent concentration ratio. It should be noted that trifluoroethanol evaporates completely during the electrospinning process, and any residual amounts are insignificant and evaporate shortly after being deposited on the collector. The residual level of trifluoroethanol was not evaluated separately because it evaporates quickly.

  1. In the introduction section, a short description of the types of calcium phosphates used in bone repair systems should be mentioned (i.e. doi.org/10.1007/s40204-015-0045-z; doi.org/10.1039/C7RA11278E; doi.org/10.3390/polym13010053) with special attention paid to these used in electrospun scaffolds (i.e. doi.org/10.3390/cryst11020199). Limited or contradictory results of the incorporation of ACP into electrospun scaffolds for bone applications should be also discussed.  

We have included the following text in the Introduction:

Due to its disordered structure and structural defects, ACP is a precursor phase of HAP that exhibits higher reactivity, solubility, and bioactivity [20]. ACP can release calcium and phosphate ions in a physiological context. This phenomenon promotes the growth of HAP crystals on the scaffold's surface and enhances the bone cell differentiation process towards osteogenesis [21].

Although few studies have investigated the integration of ACP into electrospun scaffolds for bone applications [22,23], these scaffolds have shown that they are promising for bone tissue engineering and have the potential to promote bone regeneration due to their rapid mineralization and improved cell adhesion [24–26]. The quantity and evenness of ACP dispersion in the electrospun fibers have an impact on the morphology, stability, and functional traits of the resulting scaffolds. The overabundance or deficiency of ACP could potentially compromise both the mechanical and bioactive properties of these scaffolds. In addition, the physicochemical properties of electrospun scaffolds incorporating ACP in conjunction with proteins such as collagen have not been studied, nor have their effects on primary cells been evaluated. We make note of this in our Introduction section (line 66).

  1. In the introduction, the Authors should add one more paragraph in which they “briefly mention the main aim of the work and highlight the main conclusions” as stated in Instructions for Authors.

We included the next paragraph: 

To better comprehend the effects of amorphous calcium phosphate (ACP) on the physicochemical characteristics of electrospun scaffolds made of polylactic acid (PLA) and collagen (COL), and their implications for bone regeneration, we created and analyzed the physicochemical properties, as well as evaluated the biological properties of electrospun scaffolds incorporating PLA and COL alongside hydroxyapatite (HAP) or amorphous calcium phosphate (ACP), labeled MHAP and MACP. The aim is to investigate the impact of bioceramics on scaffold properties and their subsequent influence on the behavior of hWJ-MSCs in regenerative medicine studies. We make note of this in our Introduction section (line 88).

  1. The description of hWJ-MSCs should be moved from the discussion section into the introduction because the Authors should somehow justify choosing mesenchymal stromal cells instead of osteoblasts. Additionally, the first paragraph in the discussion should be also transferred into the introduction section

We appreciate these comments. These have already been done. We make note of this in our Introduction section (line 81).

  1. The reference for glass transition, melting, and crystallization of PLA should be added. 

The requested reference was in the initial document; however, a new reference was included for this material.

  1.   Leonés, A.; Peponi, L.; Lieblich, M.; Benavente, R.; Fiori, S. In Vitro Degradation of Plasticized PLA Electrospun Fiber Mats: Morphological, Thermal and Crystalline Evolution. Polymers (Basel). 2020, 12, 2975.

  1. How was the Young’s modulus calculated? The obtained values should be described in the results section. Is the elastic limit for MACP 1.5 or 2.3 MPa? 

In response to this feedback, we have included the methodology used to determine the Young's modulus of each scaffold and improved the clarity of the value presentation to improve understanding. The following section details the process:

Young's modulus values were calculated by determining the slope of the stress-strain diagram within the elastic zone for each scaffold. The PLA scaffold showed values of 102 MPa ± 4 MPa, while the MHAP and MACP scaffolds showed a decrease with values of 34 MPa ± 7 MPa and 49 MPa ± 13 MPa, respectively. We make note of this in our Young Modulus of Scaffolds section (line 361).

  1. Based on the obtained results presented in Fig. 5, it is difficult for the reader to see the significant differences between the scaffolds mentioned by the Authors. The intensities of fluorescence should be provided. 

We have included the fluorescence intensities and the following text:

Fluorescence intensity measurements obtained using ImageJ confirm the consistency with the images obtained. In particular, MACP scaffolds had 33,801, 51,885, and 52,424 on days one, three, and five, respectively, while MHAP scaffolds had 16,814, 23,593, and 31,09. In addition, at the same time points, the PLA scaffold showed values of 15,487, 22,356, and 28,191. We make note of this in our Adhesion and proliferation cellular section (line 379).

  1. Why is such a significant reduction in cells' viability observed after the first day of incubation with all types of scaffolds? Is such a phenomenon observed in the literature? Are the scaffolds cytotoxic to cells based on ISO 10993-5 recommendations?

We thank for this opportunity to clarify this point. ISO 10993-5:2009 - Biological evaluation of medical devices describes test methods for evaluating the in vitro cytotoxicity of medical devices. The cytotoxicity test can be performed with an extract of the device or with the device in direct contact with cells. The cells must be partially confluent within 24 to 48 hours prior to culture, and are followed for 24 and 48 hours after exposure to the device in any of its forms. In our case, we did not perform a cytotoxicity assay, but rather a cell proliferation assay. We were interested in evaluating the ability of the cells to adhere, proliferate, and invade the scaffolds; therefore, the procedure consisted of seeding the cells directly onto the scaffolds and following them over time. 

Depending on the physicochemical properties of the scaffold, it is more or less difficult for the cells to adhere and proliferate, as shown in Figure 6, where the cells proliferate less on the PLA scaffold due to its hydrophobicity and more on TCP, which is the gold standard for adherent cell culture.

  1. How were the scaffolds sterilized before cytocompatibility studies? How were they placed in the wells? 

We appreciate the reviewer's question regarding the sterilization and placement of scaffolds for cytocompatibility studies. 

To provide clarity, we state in the document:

"The scaffolds were cut into circular discs 1 cm in diameter and plated in 48-well plates. The scaffold discs were then sterilized using the BIOBEAM 2000 caesium-137 gamma irradiator (Eckert & Ziegler BEBIG GmbH, Germany). Two separate doses of 25 Gray each were administered at room temperature for 15 minutes under light-protected conditions.” We make note of this in our Cell adhesion and proliferation section (line 162).

  1. In both osteogenic and immunohistochemical studies, the cells seeded without any materials should be used as a control. It is really difficult to observe any differences in Fig. 9.

Thank you for this suggestion. In the revised version of the manuscript, we have included the osteogenic differentiation controls of cells seeded in culture plates. For the osteocalcin (OCN) immunohistochemical staining, we have included the negative controls of the assay. We note this in the Materials and Methods section, line 208 and in Figures 8 and 9.

In Figure 9, we included the negative controls of the immunohistochemical staining to better see the differences between the scaffolds.

  1. The conclusions must be modified as in the presented form they are not supported by the results. In the manuscript, there is no evidence to support the claims about the gradual and regulated release of ions or about manipulating the physicochemical properties of the scaffolds. 

The following paragraph of conclusions has been drafted in the document

This study comprehensively characterizes hydroxyapatite (HAP) and amorphous calcium phosphate (ACP) and their incorporation into electrospun scaffolds, highlighting their potential for bone tissue engineering. SEM analysis reveals distinct morphological differences between HAP and ACP, with HAP exhibiting uniform particle size and smooth surface, while ACP forms irregular aggregates with rough texture. The Ca/P ratio variations are verified by EDX analysis. Successful scaffold incorporation is demonstrated by FTIR-ATR spectra, and calcium binding to phosphate groups is revealed by XPS analysis. The addition of collagen and calcium phosphate causes changes in thermal behavior, as indicated by DSC results. Young's modulus values evidence the changes in scaffold mechanics. Cellular studies showed that MACP scaffolds significantly increased adhesion, proliferation, osteogenic differentiation and growth factor secretion of hWJ-MSCs. In addition, SEM and alizarin red staining demonstrated their mineralization potential, positioning MACP scaffolds as promising candidates for bone tissue engineering. These results highlight the critical role of scaffold composition in influencing cellular behavior and tissue regeneration.

We note this in the Conclusions section (line 593).

  1. In the discussion, it is claimed that greater in vitro adhesion and proliferation of hWJ MSCs onto MACP scaffolds might be a result of modification in both the roughness and wettability of the fibers made of ACP compared with MHAP and PLA scaffolds. Such features were not measured during the study. Even in the discussion section, such statements should be confirmed by the results or by the very well-established phenomena described in the literature. 

Upon re-evaluation, we acknowledge that our previous statement regarding the factors influencing adhesion and proliferation of hWJ MSCs on MACP scaffolds did not have direct experimental evidence in our study. We apologize for any confusion and have modified the Discussion section accordingly. We now emphasize that although we did not quantitatively measure roughness and wettability, our results are consistent with established phenomena in the literature. Our observations are supported by a study by Zhao et al [25], which reported similar results regarding cell behavior on comparable scaffolds.

  1. The sentence “In addition, a uniform distribution of cells on the scaffold, matrix formation, and prominent cell extensions were observed, suggesting a favourable interaction between the cells and the scaffold” is misleading. What are these interactions? What is that matrix? The term „extracellular matrix” must be also defined because it is really hard to interpret such statements.  

We appreciate your comments and have revised the text accordingly. 

"SEM images showed cell adhesion in all scaffolds, but the inherently hydrophobic nature of the PLA scaffold resulted in a lower number of cells present (Figure 8a-l). Conversely, the MHAP scaffold showed higher cell density when cultured with osteogenic differentiation medium. Furthermore, when hWJ-MSCs were seeded onto the MACP scaffold, they formed a uniform and dense cell layer covering the electrospun fibers, with cells evenly distributed throughout the scaffold. Extracellular matrix (ECM) formation and prominent cell extensions were observed, indicating the presence of a three-dimensional network of proteins and molecules secreted by the cells into their environment. This network provides structural and biochemical support and suggests a beneficial interaction between cells and the scaffold, enabling cell adhesion and growth in the scaffold environment. [47].”

Based on your feedback, we recognize the importance of clarifying the meaning of "extracellular matrix" (ECM). The ECM is a complex, three-dimensional network of proteins and bioactive molecules produced by cells. The scaffold provides critical structural and biochemical support to cells, aiding their adhesion, proliferation and promote the synthesis and secretion of extracellular matrix components [47].

We note this in the Calcium phosphate compounds-containing scaffolds induce the osteogenic differentiation of hWJ-MSC section (line 456).

  1. The Authors mentioned that calcium phosphates may enhance the polarity and Coulomb forces and could change the viscosity of the obtained suspension. Please provide suitable references and values of viscosities in Table 1. 

We appreciate the reviewer's request for additional information regarding the impact of calcium phosphates on viscosity and relevant references. We have excluded the data on the solutions' viscosity since we do not possess the necessary values to answer this question. The revised text is as follows:

“Furthermore, imaging revealed a decrease in fiber diameter in MACP and MHAP scaffolds due to COL and Ca/Ps compounds. These materials increase polarity and Coulomb interaction forces, thereby increasing electrostatic repulsion, reducing fiber diameter, and producing high surface area scaffolds [56]. This feature helps to create a network of interconnected micropores that promote cell signaling and the transport of nutrients and factors during tissue regeneration [51].” We note this in the Discussion section (line 520)

We appreciate the reviewer's suggestion and will consider incorporating viscosity values that are relevant to our future research, particularly when rheological analysis forms a vital aspect of the investigation.

  1. The manuscript must be checked in terms of English language and typos. For example: • not all abbreviations are explained in the Abstract section; • 2,2,2,2 Trifluoroethanol; • “dihedrite”’; • 5×104 hWJ-MSCs; • Fig. 2 – “Amide I y II”; • “see Figure 7b and 97”; • Fig. 7 – “0 horas”. 

The respective corrections were made in the document and marked in green.

  1. The Authors’ contributions section is not in agreement with the Journal’s requirements. Please mention all Authors and their contributions by following the instructions for authors. 

We have made the respective correction and modified the text as follows:

“William Cárdenas-Aguazaco and Ingrid Silva-Cote participated extensively in the execution, writing, analysis, and interpretation of the results of the study. Edwin Gómez-Pachón played a crucial role as co-director of the study and in critically reviewing the final document. Adriana Lara assisted in the execution of the in vitro evaluations and in the adjustment of all figures. Bernardo Camacho was responsible for securing funding for the development of the study and for critically reviewing the final manuscript”. We note this in the Authors’ contributions section (line 608)

  1. The quality of Figures 1, 2, and 3 must be improved.   

The quality of the figures has been improved and at the same time the original images have been sent so as not to lose their quality.

We appreciate your feedback, which has improved the clarity and rigor of our manuscript. Thank you, your contributions have been invaluable in refining our work.

Reviewer 2 Report

Comment

The manuscript entitled “Electrospun Scaffolds of Polylactic Acid, Collagen, and Amorphous Calcium Phosphate for Bone Repair” is quite interesting and well-discussed. However, there are some weak points that should be addressed.

1.     The manuscript was not well prepared. There are several inconsistent places, e.g., some locations use “hour or hours”, but many places use “h”, some locations use “minutes”, but many places use “min”. Authors should check and correct wording for consistency throughout the manuscript.

2.     The authors are recommended to review all abbreviations in the Abstract and in the Materials and Methods section. The full name of an abbreviation such as SEM, PLA, HAP, ACP, COL, hWJ-MSC, MHAP, MACP, etc. should be written only when first mentioned, both in the Abstract and in context.

3.     Please check the unit of FTIR result, -1 should be in superscript.

4.     The size unit should be written only in one place. For example, 0.372 μm ± 0.196 μm should be 0.372 ± 0.196 μm or 412 nm ± 58 nm should be 412 ± 58 nm.

5.     Authors should check the space between lines or paragraphs to be in accordance with the journal format.

6.     In intracellular or cell-related studies (Sections 2.6-2.10), the authors should address both negative and positive controls.

English is fine for this manuscript but I am concerned about the consistency as mentioned in the comments.

Author Response

The manuscript entitled “Electrospun Scaffolds of Polylactic Acid, Collagen, and Amorphous Calcium Phosphate for Bone Repair” is quite interesting and well-discussed. However, there are some weak points that should be addressed.

  1. The manuscript was not well prepared. There are several inconsistent places, e.g., some locations use “hour or hours”, but many places use “h”, some locations use “minutes”, but many places use “min”. Authors should check and correct wording for consistency throughout the manuscript.

We appreciate the reviewer's attention to the consistency of terminology in our manuscript. Consistency of terminology is essential for clarity and professionalism in scientific writing. After a thorough review, we have made the necessary revisions to ensure consistency in the use of time units.

In the revised manuscript, we have standardized time units to "hours" and "minutes" for greater consistency throughout the text. We believe that these changes improve the readability and overall quality of the manuscript.

  1. The authors are recommended to review all abbreviations in the Abstract and in the Materials and Methods section. The full name of an abbreviation such as SEM, PLA, HAP, ACP, COL, hWJ-MSC, MHAP, MACP, etc. should be written only when first mentioned, both in the Abstract and in context.

We appreciate the reviewer's careful assessment of our manuscript, particularly regarding the use of abbreviations. Consistency is critical in our document, and we have carefully reviewed the manuscript to ensure that SEM, PLA, HAP, ACP, COL, hWJ-MSC, MHAP, and MACP are fully spelled when first mentioned in both the Abstract and the Materials and Methods section.

  1.     Please check the unit of FTIR result, -1 should be in superscript.

We have reviewed the document and adjusted the FTIR results to accurately represent the units. This was achieved by superscripting the "-1" to conform to standard scientific notation, ensuring precision and accuracy in our data presentation.

  1.     The size unit should be written only in one place. For example, 0.372 μm ± 0.196 μm should be 0.372 ± 0.196 μm or 412 nm ± 58 nm should be 412 ± 58 nm.

We have reviewed the manuscript and made the necessary adjustments to improve the presentation of size units. Specifically, we have removed size units from numerical values to ensure consistency throughout the manuscript.

  1.     Authors should check the space between lines or paragraphs to be in accordance with the journal format. 

Thank you for your feedback. We have ensured that the line and paragraph spacing in the manuscript follows the journal's formatting guidelines. We have made the necessary adjustments to be consistent with the journal's requirements.

  1.     In intracellular or cell-related studies (Sections 2.6-2.10), the authors should address both negative and positive controls.

Thank you for your suggestion to include both negative and positive controls in our intracellular and cell-related studies (Sections 2.6-2.10). We agree that this will provide a more comprehensive assessment of our experimental results. We include the controls to ensure that the rigor and validity of our cellular experiments are maintained. This will improve the quality of our results and contribute to the robustness of our conclusions.

Comments on the Quality of English Language

English is fine for this manuscript but I am concerned about the consistency as mentioned in the comments.

We appreciate the reviewer's contribution to enhancing our manuscript's quality.

Round 2

Reviewer 1 Report

The Authors improved the manuscript significantly. Nonetheless, a few aspects should be reconsidered before publication.

1. The Authors wrote they provided better SEM images in Fig. 1 – that is not true. Once again, I ask the Authors to change the SEM images in Fig. 1. Current images are not representative and it is really hard to observe any differences in the morphology of the particles. The description of the green arrow should be also provided in the Figure’s caption.

2. I am not convinced that the proposed ImageJ-based method of fluorescence intensity is reliable. I thought the Authors would use a plate microreader. If the plate microreader is not available, please delete the obtained results of ImageJ calculations.

3. I also cannot fully agree with the Authors that the resazurin test measures cell proliferation. In my opinion, it is a test that measures metabolic activity, thus, relates to cell viability. The same statement is given in the materials and method section by the Authors. Taking into consideration ISO standards, a reduction of cell viability by more than 30 % is considered a cytotoxic effect. In the presented work, a significant reduction in cell viability is observed after 24 h of incubation with each material – it must be commented.

4. In Fig. 9 – the captions for e, f, g, and h images are missing. Moreover, I do not see the images for PLA and control (cells incubated with osteogenic medium without scaffold).

5. The XRD data is missing so in the whole manuscript the Authors should state that obtained ACP is most likely amorphous. The statement about amorphous natures based on morphology only is not correct.

6. In the case of the control presented in Fig. 8m, it should be an image of a well with cells after alizarin addition instead of the image of cells.

7. The term “Ca/P” should be used only in the case of calcium to phosphorus ratio. As an abbreviation of calcium phosphate it should be “CaP”.

8. The supplementary file in the .eps extension is hard to open. In the presented supplementary file I do not see the legend. Pictures g) and h) are the same. The interpretation of the obtained supplementary data is missing in the main manuscript.

9. Fig. 1 – FTIR: there are still typos such as “amide I y II” and PO4-; XPS: the size of the labels on the x and y axis is too big.

10. The Authors observed that after the addition of CaP particles, the Young’s modulus of the obtained scaffolds decreased.  A short comment with references should be provided as it is not a well-known phenomenon. The maximum stress during the plastic deformation should be mentioned in the text.

11. In the conclusion section, further perspectives should be added in the last paragraph (i.e. degradation test, mineralization assay). 

The English language is understandable. 

Author Response

We appreciate all of the valuable comments from the reviewer of our work. We have revised our manuscript according to the reviewers’ comments, questions, and suggestions. We believe that the manuscript has been further improved.

  1. The Authors wrote they provided better SEM images in Fig. 1 – that is not true. Once again, I ask the Authors to change the SEM images in Fig. 1. Current images are not representative and it is really hard to observe any differences in the morphology of the particles. The description of the green arrow should be also provided in the Figure’s caption.

We appreciate the reviewer's feedback and recognize the importance of presenting clear and representative SEM images to readers. We are improving the quality of the SEM images in Figure 1. We have included higher magnification SEM images that more accurately depict the particles' morphology in hydroxyapatite (HAP) and amorphous calcium phosphate (ACP). In addition, descriptions of the green arrows have been added to the caption of the figure to enhance understanding of the SEM images.

  1. I am not convinced that the proposed ImageJ-based method of fluorescence intensity is reliable. I thought the Authors would use a plate microreader. If the plate microreader is not available, please delete the obtained results of ImageJ calculations.

We appreciate the reviewer's feedback and understand his concerns about the reliability of the ImageJ-based fluorescence intensity measurement method. We did not perform fluorescence intensity measurements on hWJ-MSC-GFP using a plate microreader. Instead, we sought to provide valuable insight into the cellular responses to our scaffolds using alternative methods such as image analysis with ImageJ. However, to address the concern raised by the reviewer, we have removed the fluorescence data obtained from software calculations from the manuscript.

  1. I also cannot fully agree with the Authors that the resazurin test measures cell proliferation. In my opinion, it is a test that measures metabolic activity, thus, relates to cell viability. The same statement is given in the materials and method section by the Authors. Taking into consideration ISO standards, a reduction of cell viability by more than 30 % is considered a cytotoxic effect. In the presented work, a significant reduction in cell viability is observed after 24 h of incubation with each material – it must be commented.

We thank the reviewer for his insightful comments and clarification of the nature of the resazurin assay, which measures metabolic activity and thus cell viability. We agree with the reviewer's interpretation and appreciate the importance of precision in our report.

Our study was designed to use the resazurin assay to investigate the effect of biomaterials, specifically collagen and calcium phosphates (CaPs), on cellular behavior and metabolic activity on the scaffolds. The study results showed differences in cell proliferation due to differences in scaffold composition and surface wetness, consistent with the reviewer's observations.

In addition, we recognize the importance of addressing the decrease in cell viability observed after 24 and 48 hours of incubation with each scaffold, given the potential for cytotoxic effects. To improve the biocompatibility evaluation and support the safety of our scaffolds for potential biomedical applications a cytotoxicity test was performed according to the ISO 10993-5 standard. We refer to this in our Materials and methods, Results and Discussion sections (Line 196, 436, 608 respectively)

We would like to thank the reviewer for his valuable input, which improved and expanded the analysis of our research results.

  1. In Fig. 9 – the captions for e, f, g, and h images are missing. Moreover, I do not see the images for PLA and control (cells incubated with osteogenic medium without scaffold).

The referenced captions were placed again in the document. The images corresponding to the controls are shown in supplementary Figure 2.

  1. The XRD data is missing so in the whole manuscript the Authors should state that obtained ACP is most likely amorphous. The statement about amorphous natures based on morphology only is not correct.

We appreciate the reviewer's careful consideration of our manuscript and the suggestion to include XRD characterization of the calcium phosphates, particularly amorphous calcium phosphate (ACP), and to explicitly state its amorphous nature.

In response to this valuable suggestion, we have performed XRD characterization and analysis of the calcium phosphates in our study. We have included this new data in our paper to provide a more thorough description of ACP.

Materials and Methods (line 120), Results (line 283), and Discussion (line  558)

  1. In the case of the control presented in Fig. 8m, it should be an image of a well with cells after alizarin addition instead of the image of cells.

The control Tissue Culture Plate (TCP) shown in Figure 8m corresponds to cells seeded on a culture plate and stained using alizarin red. No calcium deposition was observed in the hWJ-MSCs during the evaluated period. We reported these in Results  line 518

  1. The term “Ca/P” should be used only in the case of calcium to phosphorus ratio. As an abbreviation of calcium phosphate it should be “CaP”.

The document has been revised and the abbreviations for the Ca/P ratio and the calcium phosphates (CaPs) have been corrected according to the context.

  1. The supplementary file in the .eps extension is hard to open. In the presented supplementary file I do not see the legend. Pictures g) and h) are the same. The interpretation of the obtained supplementary data is missing in the main manuscript.

We have re-included the footer as requested by the reviewer. We apologize for any inconvenience caused. Furthermore, we have completed the analysis of every signal observed in the XPS.

  1. Fig. 1 – FTIR: there are still typos such as “amide I y II” and PO4-; XPS: the size of the labels on the x and y axis is too big.

The respective change was made, according to the reviewer's suggestion.

  1. The Authors observed that after the addition of CaP particles, the Young’s modulus of the obtained scaffolds decreased. A short comment with references should be provided as it is not a well-known phenomenon. The maximum stress during the plastic deformation should be mentioned in the text.

We appreciate the reviewer's comments and suggestions, which have strengthened our manuscript.

We have revised our work in response to the concerns raised and included an analysis in the paper. Our analysis specifically addresses the decrease in Young's modulus when CaP particles are added to our scaffolds, providing a more detailed explanation.

Additionally, plastic deformation is reduced to 20% for MHAP and 32% for MACP in these electrospun polymer matrices, each with an ultimate stress point of approximately 1.9 MPa and 2.4 MPa, respectively. This suggests that MHPA and MACP have lower elastic properties (Young's modulus and yield stress) compared to PLA, resulting in less mechanical resistance. Additionally, PLA has greater ductility when plastically deformed, while MHPA and MACP are more brittle. (line 408 )

The addition of CaPs to MHAP and MACP scaffolds significantly impacted their mechanical properties. This was due to decreased yield stress and Young's modulus in the scaffolds, mainly caused by CaPs aggregation. Such aggregation affected the intermolecular connections within the PLA molecules, leading to a higher incidence of mesostructural defects within the polymer. Consequently, the scaffolds' ability to elongate before reaching the point of failure was greatly reduced. (line 601)

  1. In the conclusion section, further perspectives should be added in the last paragraph (i.e. degradation test, mineralization assay).

Thank you for this suggestion. The following paragraph were included in conclusions:

“In future studies, it is important to perform mineralization tests in simulated body fluid (SBF) according to ISO 23317:2014 to evaluate the suitability of the calcium phosphates used in these scaffolds for biomedical applications. It is worth noting that the degradation of the scaffolds has not yet been conclusively determined. Our ongoing research aims to address these critical issues and elucidate the influence of calcium phosphates on the mineralization and degradation of the scaffolds for their potential use in biomedical applications.”

Round 3

Reviewer 1 Report

Dear Authors,
I am satisfied with the improvements to the manuscript you have made. The current version of your study is suitable for publication.
Regards

The English language is understandable.